# Structural insights into loss of function of a pore forming toxin and its role in pneumococcal adaptation to an intracellular lifestyle

Dilip C. Badgujar[1☯], Anjali Anil[1☯], Angharad E. Green[2☯], Manalee Vishnu Surve[1], Shilpa Madhavan[1], Alison Beckett[3], Ian A. Prior[3], Barsa K. Godsora[1], Sanket B. Patil[1], Prachi Kadam More[1], Shruti Guha Sarkar[1], Andrea Mitchell[4], Rinti Banerjee[1], Prashant S. Phale[1], Timothy J. Mitchell[4], Daniel R. Neill[2]*, Prasenjit Bhaumik[1]*, Anirban Banerjee[1]*

1 Department of Biosciences and Bioengineering, Indian Institute of Technology Bombay, Mumbai, Maharashtra, India, 2 Department of Clinical Infection, Microbiology and Immunology, Institute of Infection, Veterinary and Ecological Sciences, University of Liverpool, Liverpool, United Kingdom, 3 Division of Cellular and Molecular Physiology, Institute of Translational Medicine, University of Liverpool, Liverpool, United Kingdom, 4 Institute of Microbiology and Infection, College of Medical and Dental Sciences, University of Birmingham, Birmingham, United Kingdom

☯ These authors contributed equally to this work.
* D.Neill@liverpool.ac.uk (DRN); pbhaumik@iitb.ac.in (PB); abanerjee@iitb.ac.in (AB)

**Data Availability Statement:** All files regarding Ply-NH crystal structure are available in PDB database (accession number 6JMP). All other

## Abstract

The opportunistic pathogen *Streptococcus pneumoniae* has dual lifestyles: one of an asymptomatic colonizer in the human nasopharynx and the other of a deadly pathogen invading sterile host compartments. The latter triggers an overwhelming inflammatory response, partly driven via pore forming activity of the cholesterol dependent cytolysin (CDC), pneumolysin. Although pneumolysin-induced inflammation drives person-to-person transmission from nasopharynx, the primary reservoir for pneumococcus, it also contributes to high mortality rates, creating a bottleneck that hampers widespread bacterial dissemination, thus acting as a double-edged sword. Serotype 1 ST306, a widespread pneumococcal clone, harbours a non-hemolytic variant of pneumolysin (Ply-NH). Performing crystal structure analysis of Ply-NH, we identified Y150H and T172I as key substitutions responsible for loss of its pore forming activity. We uncovered a novel inter-molecular cation-π interaction, governing formation of the transmembrane β-hairpins (TMH) in the pore state of Ply, which can be extended to other CDCs. H150 in Ply-NH disrupts this interaction, while I172 provides structural rigidity to domain-3, through hydrophobic interactions, inhibiting TMH formation. Loss of pore forming activity enabled improved cellular invasion and autophagy evasion, promoting an atypical intracellular lifestyle for pneumococcus, a finding that was corroborated in *in vivo* infection models. Attenuation of inflammatory responses and tissue damage promoted tolerance of Ply-NH-expressing pneumococcus in the lower respiratory tract. Adoption of this altered lifestyle may be necessary for ST306 due to its limited nasopharyngeal carriage, with Ply-NH, aided partly by loss of its pore forming ability, facilitating a benign association of SPN in an alternative, intracellular host niche.

relevant data are within the manuscript and its Supporting Information files.

**Funding:** DRN and AEG acknowledge funding from a Wellcome Trust and Royal Society Sir Henry Dale Fellowship awarded to DRN (Grant No. 204457/Z/16/Z). PB acknowledges financial support from Ramalingaswami Re-entry Fellowship (Dept. of Biotechnology, Govt. of India, Grant No. BT/RLF/Re-entry/42/2011) and research grant from Dept. of Science & Technology, Govt. of India (Grant No. EMR/2016/006067). AB acknowledges research funding from Council of Scientific & Industrial Research, Govt. of India (Grant No. 27/0331/18/EMR-II) and Science and Engineering Research Board, Govt. of India (Grant No. EMR/2016/005909). The funders had no role in study design, data collection and analysis, decision to publish, or preparation of the manuscript.

**Competing interests:** The authors have declared that no competing interests exist.

## Author summary

*Streptococcus pneumoniae*, the main causative agent of pneumonia, triggers inflammation and tissue damage by producing a pore-forming toxin, pneumolysin (Ply). Ply-induced inflammation drives pneumococcal transmission from nasopharynx (its primary reservoir), but also contributes to host mortality, limiting its occupiable habitats. Here, we uncovered the structural basis for loss of pore-forming activity of a Ply variant, present in Serotype 1 ST306, and observed that this enabled adoption of an intracellular lifestyle, attenuating inflammatory responses and prolonging host tolerance of pneumococcus in the lower airways. This commensal-like lifestyle, resembling that of members of the mitis group of Streptococci, might have evolved within ST306 by loss of function *ply* mutations, compensating for limited nasopharyngeal carriage capacity by facilitating adaptation to an alternate niche.

## Introduction

Virulence of a microbe is a highly dynamic trait, dependent primarily on the niche it occupies and the availability of an alternate reservoir. The induction of excess morbidity and mortality is a cost associated with virulence for pathogens that rely upon their hosts for onward transmission. Recent studies suggest that pathogens can undergo attenuation of virulence inside the host, as long as transmission remains uncompromised [1]. Such traits enable microbial colonization and proliferation, with minimal harm to the host, and form the basis of the "infection tolerance" concept that challenges the paradigm of the arms race in infection biology [2].

*Streptococcus pneumoniae* (the pneumococcus or SPN) is a Gram positive, alpha-hemolytic bacterium and is the leading cause of community-acquired pneumonia, pediatric empyema and bacterial meningitis. SPN has a characteristic asymptomatic colonization phase in the human nasopharynx, its predominant niche and the primary reservoir for onward transmission, but can act as an opportunistic pathogen within other host sites [3]. Nasopharyngeal colonization is a prerequisite for the development of pneumococcal disease, but the relative invasiveness of SPN varies between serotypes [4], 100 different types of which have been classified, based on the composition of capsular polysaccharide [5,6]. Although pneumococcal virulence factors induce cytotoxicity, the host inflammatory response triggered against these factors is the major mediator of pathology and lethality associated with invasive pneumococcal disease (IPD). The key trigger of this dysregulated host inflammation is pneumolysin (Ply), a pore forming toxin belonging to the family of cholesterol dependent cytolysins (CDC) (S1 Fig), produced by all SPN strains. In addition to the extensive cellular damage resulting from pore forming activity on host cell membranes, Ply modulates host inflammatory and immune responses. At low levels, Ply can stimulate tolerogenic host responses via interaction with mannose receptor C type 1 (MRC-1) [7], but at the high concentrations achieved during IPD, this results in excessive inflammation, driven via its interactions with Toll-like receptors (TLR4) [8] and activation of the NOD-like receptor family pyrin domain containing 3 (NLRP3) inflammasome [9]. Ply also induces necroptosis of respiratory epithelial cells and this mode of cell death has been linked to release of IL-1α from host cells [10,11]. This IL-1 signaling has been identified to play a key role in inflammatory clearance of SPN from the nasopharynx [12] with Ply-deficient strain exhibiting slower clearance than their wild type counterparts [13]. On the other hand, inflammation triggered by the pore-forming activity of Ply in the upper respiratory tract was found to be vital for pneumococcal shedding and transmission [14]. Thus, Ply is a versatile factor, the activities of which can have different (and sometimes opposing)

consequences depending on the concentration or niche in which it is produced, variously aiding or hindering pneumococcal fitness within the host. The critical contribution of Ply to stimulation of the inflammation required to achieve transmission provides an explanation for why SPN lineages have not lost this toxin over the course of evolution, despite its potential to kill the host, leading to loss of reservoir.

Although all SPN isolates produce Ply, around 20 allelic variants with differences in their hemolytic (cytolytic) activity have been identified to date (S2 Fig) [15–17], and these polymorphisms are especially prevalent amongst serotype 1 lineages. Particularly noteworthy is the presence of a non-hemolytic variant (Ply-NH; encoded by allele 5) in serotype 1 ST306, one of the most successful pneumococcal clonal clusters [18]. ST306 is commonly associated with outbreaks of non-lethal respiratory tract infections [15]. Presence of Ply-NH is believed to be one of the key factors behind the clonal expansion of ST306 isolates [15], although it is unclear what advantage SPN derives from loss of hemolytic activity of Ply. Intriguingly, serotype 1 isolates are rarely found colonizing the nasopharynx [19]. Hence, it is unclear as to how the non-hemolytic clones have been successfully maintained in the population, despite apparent loss of their primary reservoir and the phenotype (Ply pore-induced inflammation) that is critical for host to host transmission.

Ply-NH differs from wild type hemolytic Ply (Ply-H) by the presence of 4 substitutions (Y150H, T172I, K224R, A265S) and 2 deletions (ΔV270 and ΔK271) [15] (S3 Fig). The structures of monomeric and pore forms of Ply-H were recently solved; however, none of the delineated amino acid residues critical for pore formation match with the mutations present in Ply-NH [20–22]. A detailed structural analysis of Ply-NH may therefore uncover the key interactions required for pore formation by Ply, or by CDCs in general.

In the present study, we demonstrate that although Ply-NH is able to bind and oligomerize on cholesterol containing membranes, it is unable to form pores. Our studies uncovered a novel cation-π interaction, governing pore formation in Ply-H and related CDCs, loss of which inhibits transmembrane β-hairpin (TMH) formation and subsequent pore formation in Ply-NH. Enhanced cellular invasion and evasion of host intracellular defenses, owing to loss of Ply's pore-forming ability, allows SPN to establish a novel intracellular niche in the lungs. The resulting subdued inflammatory response enables persistence of SPN in the lower respiratory tract. Extended maintenance of an active intracellular reservoir, in the absence of damaging inflammation, may prolong the period for potential transmission and successful clonal expansion of ST306.

## Results

### Ply-NH can bind and oligomerize on membranes but is incapable of forming pores

We first compared the pore formation dynamics of Ply-NH and Ply-H. Hemolysis assay performed using purified recombinant Ply (rPly) confirmed that Ply-NH remained non-hemolytic over a wide range of concentrations (up to 0.1 μM), in contrast to Ply-H, which caused complete lysis of red blood cells (RBCs) with as little as 0.01 μM protein (Fig 1A). However, pore forming ability and hemolytic activity are not always synonymous, as lack of hemolytic activity indicates the inability of the toxin to form large enough pores for the release of hemoglobin, but does not exclude the possibility that the toxin forms smaller pores. To explore this, we performed liposome leakage assays using the small fluorescent dye calcein (0.6 kDa, hydrodynamic radii ~ 0.74 nm), encapsulated at quenching concentration in liposomes. Treatment of these liposomes with increasing concentrations of Ply-H showed a time dependent increase in the fluorescence intensity associated with liposome permeabilization and release of calcein.

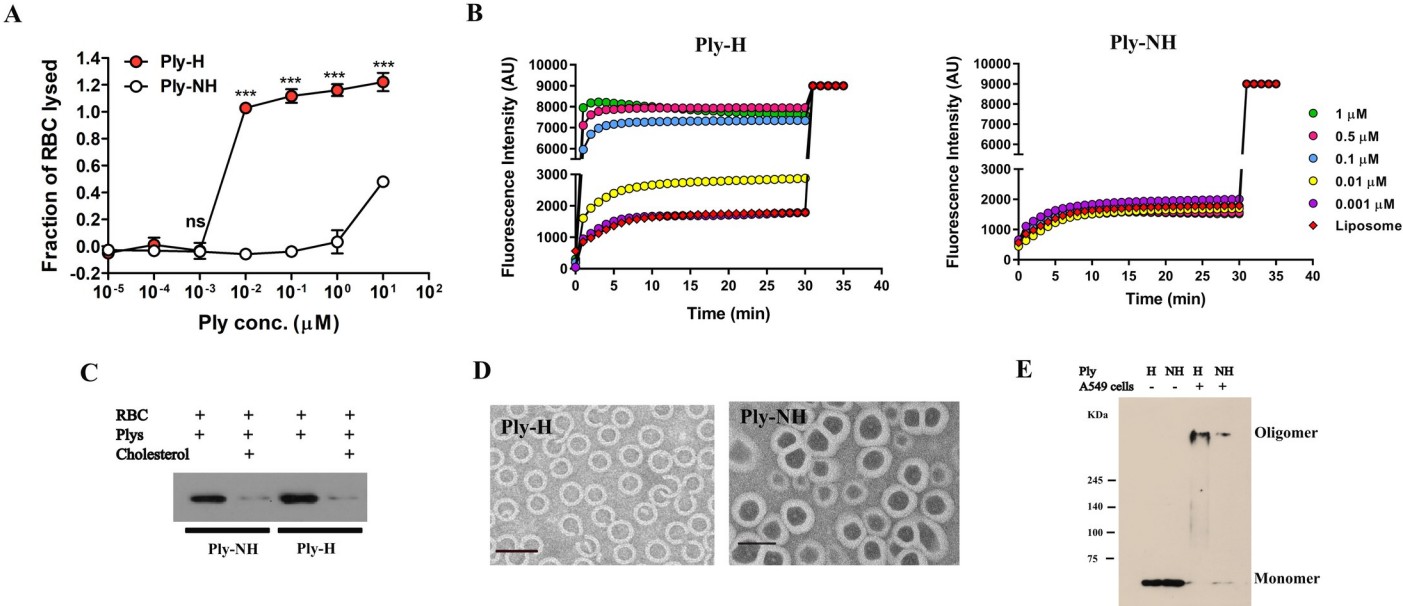

**Fig 1. Ply-NH can bind and oligomerize on membranes but is incapable of forming pores.** (**A**) Hemolysis of sheep RBCs by purified recombinant Ply-H and Ply-NH at different concentrations. (**B**) Kinetics of calcein (hydrodynamic radii ~ 0.74 nm) release from liposomes upon addition of increasing concentrations of Ply-H and Ply-NH indicating the inability of Ply-NH to form pores on membranes. (**C**) Western blot analysis of Ply-H and Ply-NH (0.01 μM) following incubation with RBC ghost membranes in the presence of cholesterol (50 μM) revealing cholesterol binding ability of both Ply-H and Ply-NH. (**D**) Transmission electron micrograph (TEM) showing assembly and oligomerization of Ply-H and Ply-NH on liposomes. Scale bar: 50 nm. (**E**) Visualization of Ply-H and Ply-NH oligomers on A549 cells using SDS-PAGE (5%) followed by immuno-blotting. Higher order oligomers and monomers were observed in samples of A549 cells ($10^6$) treated with either Ply-H or Ply-NH (0.5 μg). Purified Ply-H and Ply-NH without cells were observed as monomers only (53 kDa). Data information: Experiments are performed thrice and data of representative experiments are presented as mean ± SD of triplicate wells (A). Statistical analysis was performed using Student's two-tailed unpaired t-test (A). ns, non-significant; ***$p < 0.001$.

In contrast, Ply-NH failed to cause calcein release, confirming its inability to form even smaller pores (Fig 1B).

To determine the molecular basis for the inability of Ply-NH to form pores, each step of pore formation, namely, (a) binding of monomers to membrane cholesterol, (b) oligomerization to form pre-pore and (c) pre-pore to pore transition (TMH formation) was examined. Comparison of the cholesterol binding ability of Ply-NH with that of Ply-H was performed by pre-incubating Ply variants with cholesterol, followed by analyzing its interaction with RBC ghost membrane. Western blot analysis showed that Ply-NH interacts with the membrane in a similar manner to Ply-H and the interaction of both proteins with membranes was reduced upon cholesterol pre-treatment, suggesting that the mutations in Ply-NH do not alter its ability to bind cholesterol-containing membranes (Fig 1C). Transmission electron microscopy, used to visualize Ply assembly on membranes, demonstrated that Ply-H predominantly formed oligomeric rings (Fig 1D). Interestingly, Ply-NH also formed rings of similar size to Ply-H, which was further confirmed by SDS-PAGE, following treatment of eukaryotic membranes with the Ply variants (Fig 1E).

## Crystal structure of Ply-NH

To decipher the molecular reasons behind loss of pore forming ability in Ply-NH, the crystal structure was solved (2.2 Å resolution) and was found to consist of 4 distinct domains, D1-D4 (Figs 2A and S4 and S1 Table). This is the first reported structure of a non-pore forming CDC. D1 is present at the N-terminal region and comprises of 6 anti-parallel β strands, loops and 5 α

helices. D2 consists of five-stranded anti-parallel β sheets, which form the backbone of the structure and connect D4 with D1. D3 consists of a single antiparallel β-sheet with two α helices on either side. D4, which is connected to D2 via the Arg-Asn-Gly flexible linker, is comprised of two anti-parallel β sheets, with the conserved undecapeptide at the end of the loop, which is required for binding to cholesterol.

The overall structural fold of Ply-NH was found to be similar to that of other reported CDCs, such as anthrolysin (3CQF) [23], intermedilysin (4BIK) [24], perfringolysin (PFO; 1PFO) [25] and pneumolysin (Ply-H; 5CR6 and 4QQA) [20,26]. Superposition of Ply-NH structure with the recently reported structures of Ply-H (5CR6 and 4QQA) [20,26] produced root mean square deviation (r.m.s.d.) of 2.4 and 1.2 Å, respectively, over 471 Cα atoms. Superposition of specific domains, D1-3 of Ply-NH and Ply-H (5CR6) yielded r.m.s.d. of 0.75 Å and alignment of only D4 yielded r.m.s.d. of 0.23 Å. Comparison with one Ply-H structure, 5CR6, showed higher r.m.s.d. compared to the other (4QQA) because of a relative 10 Å movement of D4 with respect to the rest of the molecule.

Monomers in the Ply-NH crystal were found to be tightly packed and this crystallographic arrangement resembles the monomer-monomer interaction interface that may form during formation of the pre-pore complex. Although both sides of the monomer show charge complementarities, the overall surface of the structure is highly electronegative (Fig 2B). However, D3 did not show any charge complementarity. In the well-studied and closely related CDC perfringolysin of *Clostridium perfringens*, the α helices present in D3 are reported to undergo a conformational change to form TMH1 and TMH2, thereby inserting into the membrane [27]. Since most of the substitutions and deletions are present in the D3 domain of Ply-NH, we hypothesized that its non-hemolytic nature might be a consequence of the conformational changes associated with these mutations.

## Loss of a novel, essential cation-π interaction inhibits TMH formation in Ply-NH

Alignment of protein sequences indicate that the tyrosine residue at position 150 in Ply-H (Y150) is highly conserved across the different CDCs (Fig 2C). Generally, binding of CDC monomers on membranes drives conformational changes in D3 that trigger rotation of β5 away from β4, exposing the latter for interaction with β1 of an adjacent monomer [27]. The resultant π-π stacking interaction between the conserved tyrosine (in β1 of one monomer) and a semi-conserved phenylalanine (in β4 of the adjacent monomer) stabilizes the β strands of the monomers, facilitating their insertion into the membrane. Surprisingly, the phenylalanine residue (corresponding to F318 of PFO) is replaced by V287 in Ply-H (Fig 2C), suggesting that association of β1 with β4 in Ply-H might be driven by interactions other than the typically conserved π-π interactions. Mutation of three potential pairing residues of Y150, in the β4 region of Ply-H (E286, V287 and K288) to alanine resulted in loss of hemolytic activity, with the K288A mutant showing maximum loss (Figs 2D and S5A), implying its involvement in a novel cation-π interaction with Y150, essential for pore formation. This Y150-K288 interaction between adjacent monomers could also be identified in the pore-form model we developed based on the Ply-H pore complex cryo-EM structure [22] (Fig 2E). This is the first report implying cation-π interactions in pore formation by CDCs and could be relevant to a subgroup including mitilysin, which does not have the conserved phenylalanine and harbors a lysine at a similar position in β4 (S5B Fig). Structural analysis of Ply-NH revealed that the Y150H substitution leads to repulsion between two positively charged residues (H150 and K286), which may destabilize Ply-NH's pre-pore state, leading to loss of its pore forming ability (Fig 2E).

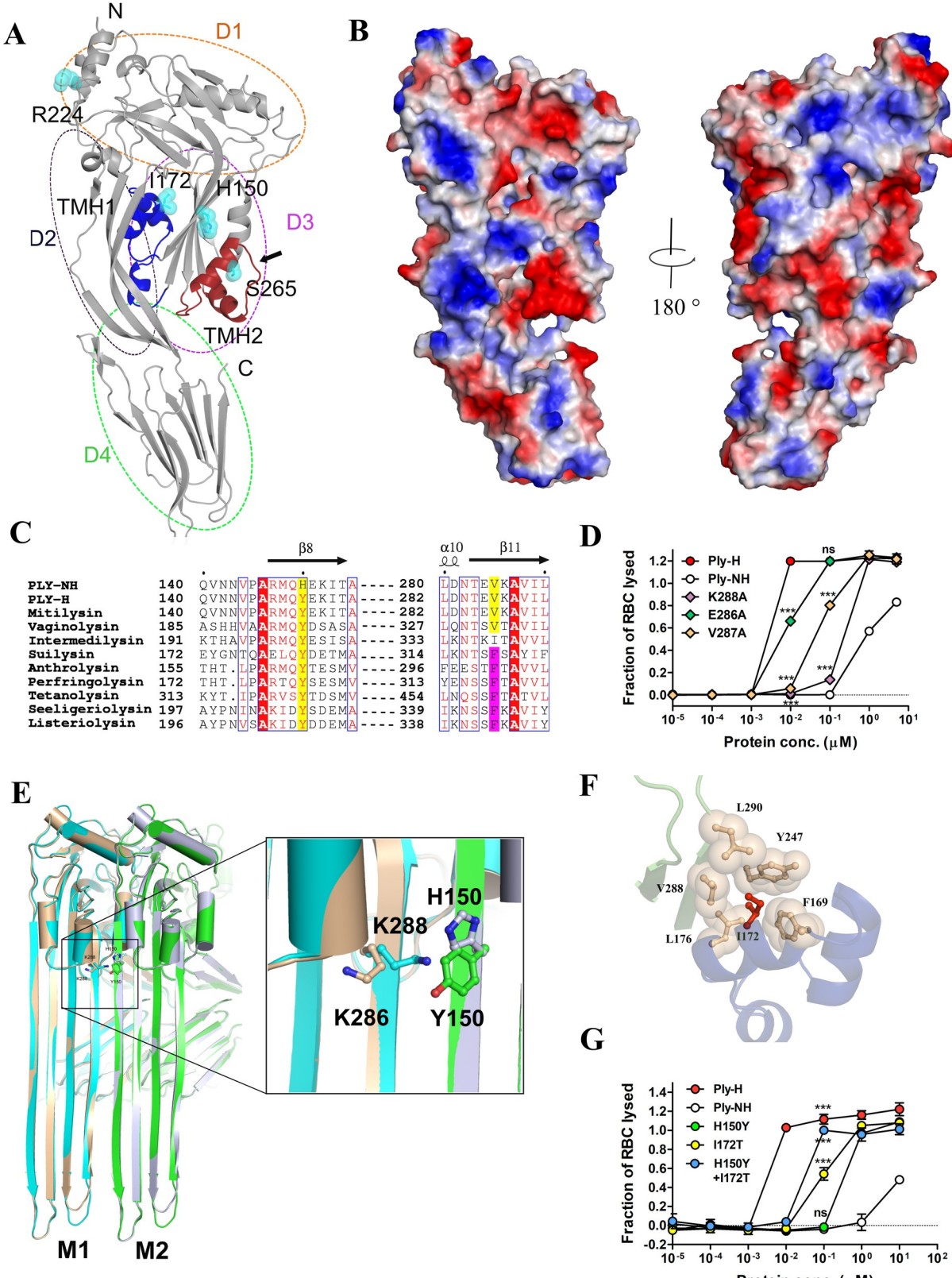

**Fig 2. Loss of an essential cation-π interaction inhibits transmembrane β-hairpin (TMH) formation in Ply-NH.** (**A**) Cartoon representation of Ply-NH crystal structure where individual domains are labelled as D1, D2, D3 and D4. The substitutions are represented as spheres along with ball and stick and colored according to elements (C-cyan, N-blue and O-red). The deletion is marked by black arrow. The TMH1 and TMH2 are also highlighted in blue and brown, respectively. (**B**) Electrostatic surface representation of Ply-NH. The blue and red indicate the electropositive and electronegative regions, respectively. (**C**) Multiple sequence alignment of CDCs highlighting the conservation

of tyrosine (highlighted in yellow, Y150 in Ply-H) and phenyl alanine (highlighted in magenta, F318 in perfringolysin). (**D**) Hemolytic activity of Ply-H and its mutated variants (E286A, V287A and K288A) at different concentrations for identification of the potential β4 residue which pairs with Y150 from β1 of neighbouring monomer. (**E**) Cartoon representation of extended TMHs of two Ply-H/Ply-NH monomers (M1 and M2 colored cyan and green for Ply-H and brown and grey for Ply-NH, respectively) in the oligomerized state. Inset: Zoomed in view showing inter-molecular cation-π interaction between K288 and Y150 in Ply-H and its disruption in Ply-NH with H150 substitution. (**F**) Close up view of the hydrophobic pocket in domain-3 formed by F169, I172, L176, Y247, V288 and L290 residues, which are shown as both sphere and ball and stick style (wheat color). The I172 (ball and stick model in red color) is found to be stabilized in the hydrophobic pocket. (**G**) Concentration dependent hemolytic profile of Ply-NH and its mutants showing gain of hemolytic activity. Individual mutations H150Y and I172T in Ply-NH show some gain of hemolytic activity, notably the double mutant (H150Y+I172T) shows gain of most of the hemolytic activity. Data information: Experiments are performed thrice and data of representative experiments are presented as mean ± SD of triplicate wells (D, G). Statistical analysis was performed using one-way ANOVA with Tukey's multiple comparison test vs Ply-H (D) or Ply-NH (G). ns, non-significant; ***$p < 0.001$.

Another important substitution in Ply-NH is threonine to isoleucine at position 172. The I172 in Ply-NH is well defined in the electron density (S4 Fig) and is found to be located at the tip of TMH1 (Fig 2F). Presence of the polar side chain of T172 in the non-polar pocket of Ply-H likely makes this region quite unstable, allowing the smooth disengagement from β3 and β4 upon binding of this toxin to the membrane. On the contrary, I172 in Ply-NH is found to be stabilized through the hydrophobic interactions involving the side chains of F169, L176, Y247, V288 and L290 (Fig 2F). The interactions from Y247, V288 and L290 might be essential to prevent the disengagement of these α helices joining β3 and β4, to form TMHs for membrane insertion and subsequent pore formation.

We next mutated H150 and I172 back to their respective Ply-H residues (H150Y and I172T) in the Ply-NH background. Though individual mutations showed some gain of activity, the double mutant (Ply-NH$^{H150Y+I172T}$) regained most of the hemolytic activity (Fig 2G). This clearly implicates Y150H and T172I as the major mutations responsible for the loss of pore forming ability of Ply-NH, by preventing disengagement of α helices joining β3 and β4 and subsequent TMH formation.

## Ply-NH is unable to form transmembrane β-hairpins (TMH1 and TMH2)

To evaluate this predicted loss of conformational dynamics, TMH1 and TMH2 formation in both Ply-H and Ply-NH was monitored using the environment sensitive fluorescent dye $N$, $N'$-dimethyl-$N$-(iodoacetyl)-$N'$-(7-nitrobenz-2-oxa-1,3-diazol-4-yl) ethylenediamine 7 (NBD). The fluorescence of NBD-labelled protein remains quenched in water but increases in a non-polar environment, thereby acting as an indicator of membrane insertion of CDCs. Two residues per TMH in Ply variants were selected, based on the structural comparison with PFO (1PFO), to monitor TMH1 and TMH2 formation. The NBD labelled monomers of Ply-H derivatives: S167C and H184C (TMH1 region) and D257C and E260C (TMH2 region), which retained their hemolytic properties (S2 Table), showed basal fluorescence emission, but exhibited significant increase in fluorescence intensity when incubated with cholesterol containing liposomes (Fig 3A–3C). NBD-labelled Ply-NH derivatives: S175C and H184C (TMH1) and D257C and E260C (TMH2), however, failed to do so (Fig 3D–3F), confirming that the inability of Ply-NH to form pores is primarily due to the structural rigidity of D3, restricting the conformational change required for the formation of TMHs.

## Abrogation of pore forming ability confers improved internalization of SPN into host cells

To compare the interaction of SPN strains harboring Ply variants with host cells *in-vitro*, we first performed invasion assays in A549 alveolar epithelial cells, using the non-encapsulated

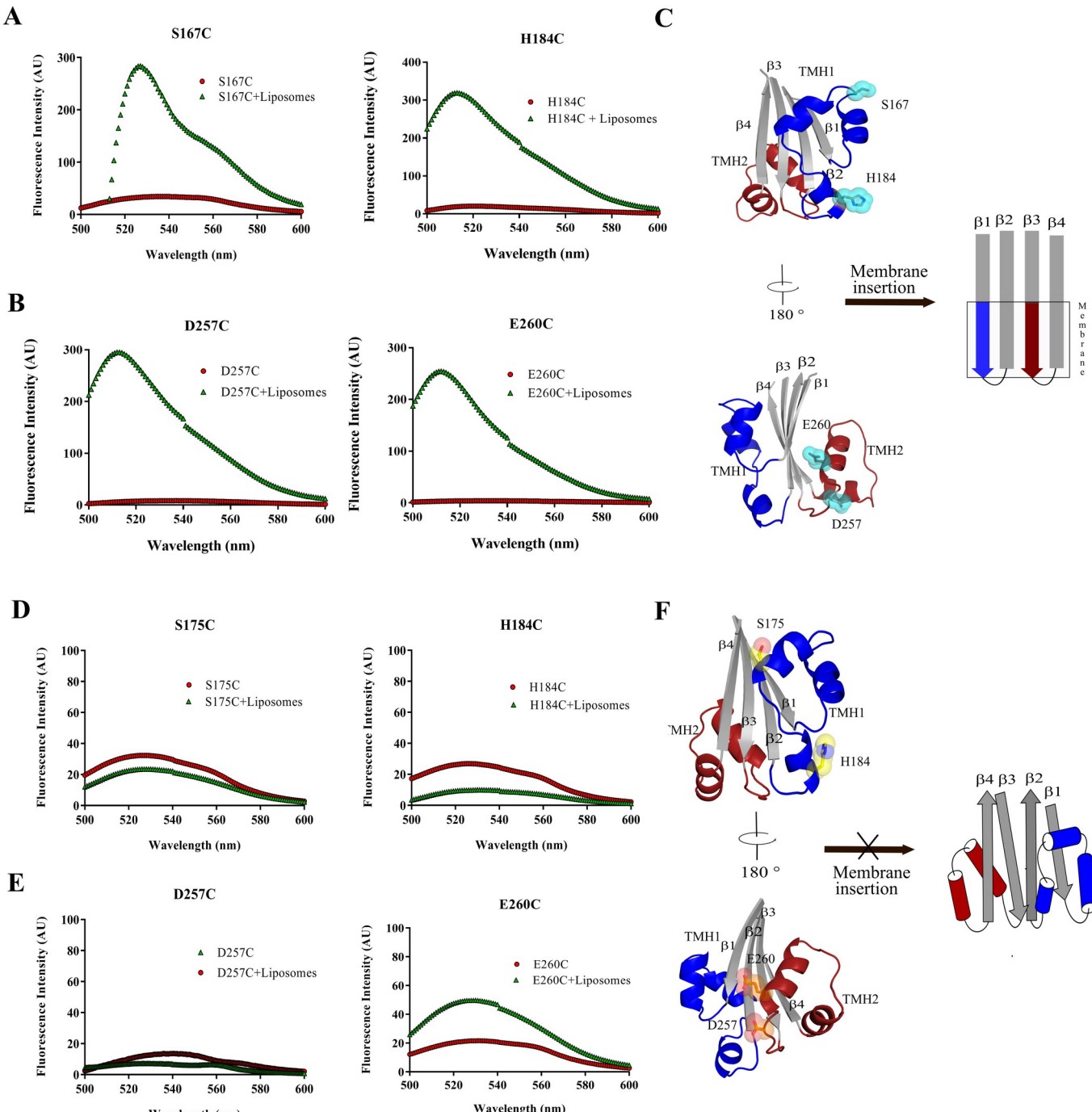

**Fig 3. Ply-NH is unable to form transmembrane β-hairpins (TMH1 and TMH2) Fluorescence intensities of all the labelled proteins with and without liposomes are depicted by green triangles and red circles, respectively.** The fluorescence emission scans of Ply-H labelled variants (**A**) S167C, H184C (TMH1) and (**B**) D257C, E260C (TMH2) and Ply-NH labelled variants (**D**) S175C, H184C (TMH1) and (**E**) D257C, E260C (TMH2) are shown. (**C,F**) Schematic representing formation of TMH in Ply-H (C) and its inability in Ply-NH (F).

derivative of serotype 2 strain D39, R6:Ply-H, that is more readily internalized than its parent strain, and its allele 5 *ply* mutant R6:Ply-NH (S6A, S6B and S7A Figs). Our findings revealed a significantly improved internalization capability of R6:Ply-NH into lung epithelial cells

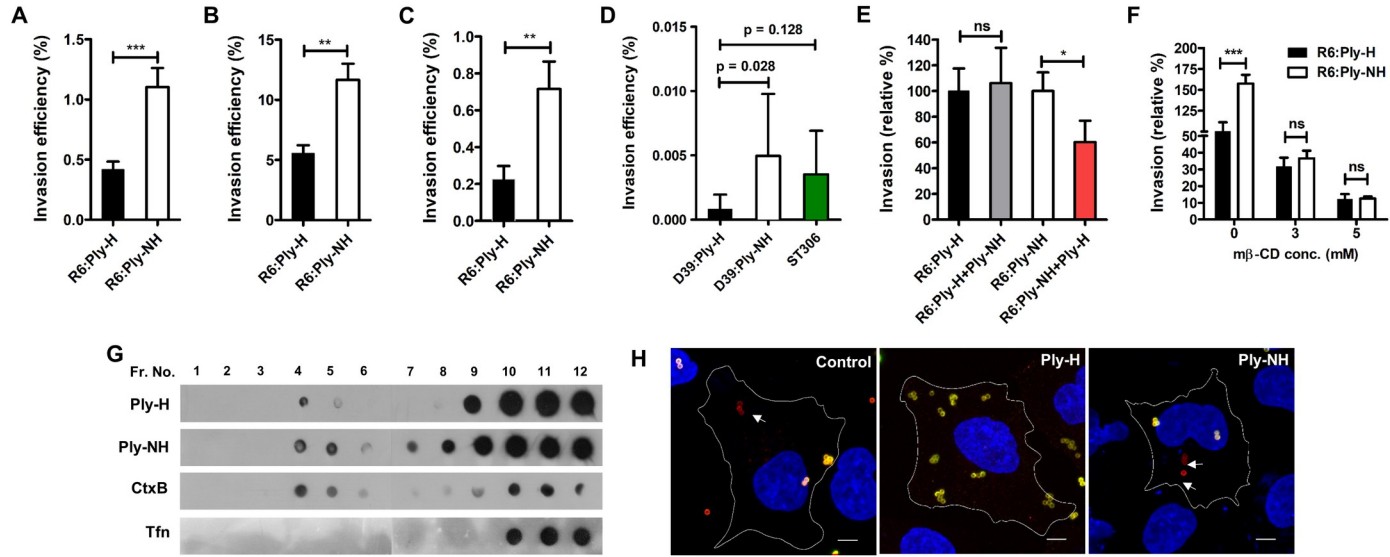

**Fig 4. Abrogation of pore forming ability confers improved internalization capability in SPN strains harbouring Ply-NH.** (**A-B**) Invasion efficiency of R6:Ply-H and R6:Ply-NH strains in A549 cells (**A**) and THP-1 cells (**B**). (**C-D**) Invasion efficiency of R6:Ply-H, R6:Ply-NH (**C**) and D39:Ply-H, D39:Ply-NH and ST306 strains (**D**) in primary human pulmonary alveolar epithelial cells. (**E**) Comparison of invasion efficiency of R6:Ply-H and R6:Ply-NH following pre-treatment of A549 cells with purified recombinant Ply-NH and Ply-H (0.1 μg/ml), respectively. (**F**) Inhibition of internalization of R6:Ply-NH following pre-treatment of A549 cells with methyl β-cyclodextrin (Mβ-CD, 3 and 5 mM). (**G**) Dot blot showing localization of Ply-H and Ply-NH in low density lipid raft fractions of A549 cell membrane. CtxB and transferrin (Tfn) was used as positive and negative control, respectively. (**H**) Immunofluorescence image showing uptake of CtxB-FITC coated latex beads (1.1 μm) by A549 cells following pre-treatment with Ply-H and Ply-NH. Internalized beads are shown in red (arrow mark) while external beads are dual (yellow) colored. Scale bar: 5 μm. Data information: Experiments are performed thrice and data of representative experiments are presented as mean ± SD of triplicate wells. Statistical analysis was performed using Student's two-tailed unpaired t-test (A-C; E-F) and one-way ANOVA with Tukey's multiple comparison test (D). ns, non-significant; *$p<0.05$; **$p<0.01$; ***$p<0.001$.

compared to R6:Ply-H (Fig 4A). Similar increased invasion efficiency was also observed in macrophages (THP-1) and primary human pulmonary alveolar epithelial cells (HPAEpiC) (Fig 4B and 4C). HPAEpiCs were also used for testing the invasive ability of encapsulated strain D39:Ply-H and the allele exchange mutant D39:Ply-NH, wherein the latter exhibited improved internalization, consistent with the previous results (Fig 4D). ST306 strain 01–1956 was also demonstrated to invade these primary cells (Fig 4D). These findings were not explained by differential Ply production in the three strains, as all produced comparable Ply per capita (S6C Fig). To examine whether enhanced internalization was conferred by Ply-NH or was due to loss of pore forming ability of Ply-NH, bacterial invasion assays were repeated following pre-treatment of A549 cells with a sub-cytotoxic concentration of rPly (S7B Fig). We observed a significant loss in internalization of R6:Ply-NH in A549 cells following pre-treat-ment with rPly-H, suggesting a role of the pore forming ability of Ply in suppression of host cell endocytic pathways (Fig 4E). Ply-H is known to segregate to lipid rafts [28], and owing to its pore forming ability, we hypothesized that it might disrupt the lipid raft mediated endocytic pathways, thereby explaining higher internalization capability of strains harboring Ply-NH. Indeed, treatment of cells with methyl β-cyclodextrin (Mβ-CD), an inhibitor of lipid raft medi-ated endocytosis, neutralized the improved internalization of R6:Ply-NH (Figs 4F and S8A). Next, we checked the segregation of Ply-NH with the low density lipid raft fraction isolated from A549 membranes, following sucrose density gradient centrifugation. Although both Ply-H and Ply-NH were observed to localize to lipid rafts (Fig 4G), only treatment with Ply-H interfered with the uptake of lipid raft endocytic pathway specific cargo cholera toxin B conju-gated latex beads by A549 cells (Figs 4H and S8B). This implies that the pore forming activity of Ply interferes with lipid raft mediated entry of SPN into eukaryotic cells, whereas its loss

facilitates such processes, conferring improved internalization capability to strains harboring Ply-NH.

## Loss of pore forming ability facilitates prolonged intracellular persistence of SPN

In order to track the fate of SPN strains harboring Ply variants, following entry into host cells, we performed penicillin-gentamicin protection assays using R6:Ply-H, R6:Ply-NH and R6:Ply-DM (a recombinant R6 strain where Ply-H is replaced with the Ply-NH$^{H150Y+I172T}$ allele which contains reversion mutations from Ply-NH in the form of H150Y and I172T) in A549 cells. Our findings demonstrate significantly improved survival of R6:Ply-NH compared to R6:Ply-H at all time points, whereas R6:Ply-DM behaved similarly to R6:Ply-H, indicating that loss of pore forming ability is beneficial for prolonged intracellular persistence of SPN (Fig 5A). Ability of R6:Ply-NH to persist longer than R6:Ply-H was also observed inside THP-1 macrophages (Fig 5B).

Damage to bacteria-containing endosomal membranes by pore forming agents is known to trigger autophagic killing, following recruitment of cytosolic "eat me" signals such as galectin-8 (Gal8) and ubiquitin (Ubq) [29,30]. Immunofluorescence analysis of A549 cells (Fig 5C), following infection with SPN, revealed that although R6:Ply-H exhibited a high degree of association with Gal8 (26.78 ± 3.17% at 10 h post infection (h.p.i)) (Fig 5D), R6:Ply-NH failed to co-localize with Gal8 at all the time points, confirming their residence inside intact vacuoles. Ubiquitination of R6:Ply-NH was also observed to be negligible compared to a high degree of association observed with R6:Ply-H (29.92 ± 0.92% at 10 h.p.i) (Fig 5E). Moreover, the majority of the Gal8 and ubiquitin positive R6:Ply-H (69.55 ± 2.81% and 65.10 ± 2.25%, respectively) associated with the autophagy marker LC3, implying targeting of R6:Ply-H towards autophagic killing (Fig 5F and 5G). Indeed, R6:Ply-H demonstrated a significantly higher colocalization with Lysotracker (44.19 ± 1.75), a dye that stains acidic compartments including lysosomes, compared to R6:Ply-NH (30.45 ± 1.02), at 18 h.p.i (Fig 5H). Although a Ply knockout mutant (R6Δ*ply*) similarly failed to localize with Gal8 or ubiquitin, a strain expressing a non-hemolytic Ply mutant, R6:Ply$^{W433F}$, colocalized with Gal8 (9.71 ± 0.41%) and ubiquitin (17.26 ± 0.64%), albeit to a lesser extent than R6:Ply-H (22.81 ± 0.74% for Gal8 and 36.58 ± 5.23% for Ubq), at 18 h.p.i (S9A–S9C Fig). This was also reflected in their intracellular survival capabilities, wherein R6Δ*ply*, but not R6:Ply$^{W433F}$, demonstrated improved survival in A549 cells relative to R6:Ply-H (S9D and S9E Fig). The association of R6:Ply$^{W433F}$ with Gal8 has also been observed in hBMECs [31], and underscores the fact that loss of hemolytic activity is not synonymous with loss of pore forming ability [32,33]. Overall, our results suggest that abrogation of Ply's pore forming ability not only confers improved internalization but also ensures safe residence of SPN inside host cells for prolonged periods, a switch in lifestyle that may be necessary for evasion from host immune mechanisms.

## Expression of Ply-NH attenuates virulence and allows host tolerance of SPN in the lower respiratory tract

Since SPN is typically considered an extracellular pathogen, we sought to determine the significance of loss of pore forming ability and capacity to adopt an intracellular niche in an *in vivo* infection scenario. We performed survival experiments in a mouse model of pneumonia with encapsulated D39 carrying a fully hemolytic pneumolysin (D39:Ply-H) and ST306 strain 01–1956, carrying Ply-NH. Survival experiments revealed a striking difference in the outcome of infection, with 90% of D39:Ply-H infected mice succumbing to infection within 48 h while only 10% of ST306 infected mice succumbed during the seven days of the experiment (Fig 6A). Furthermore, whilst D39:Ply-H infected mice quickly developed visible signs of disease

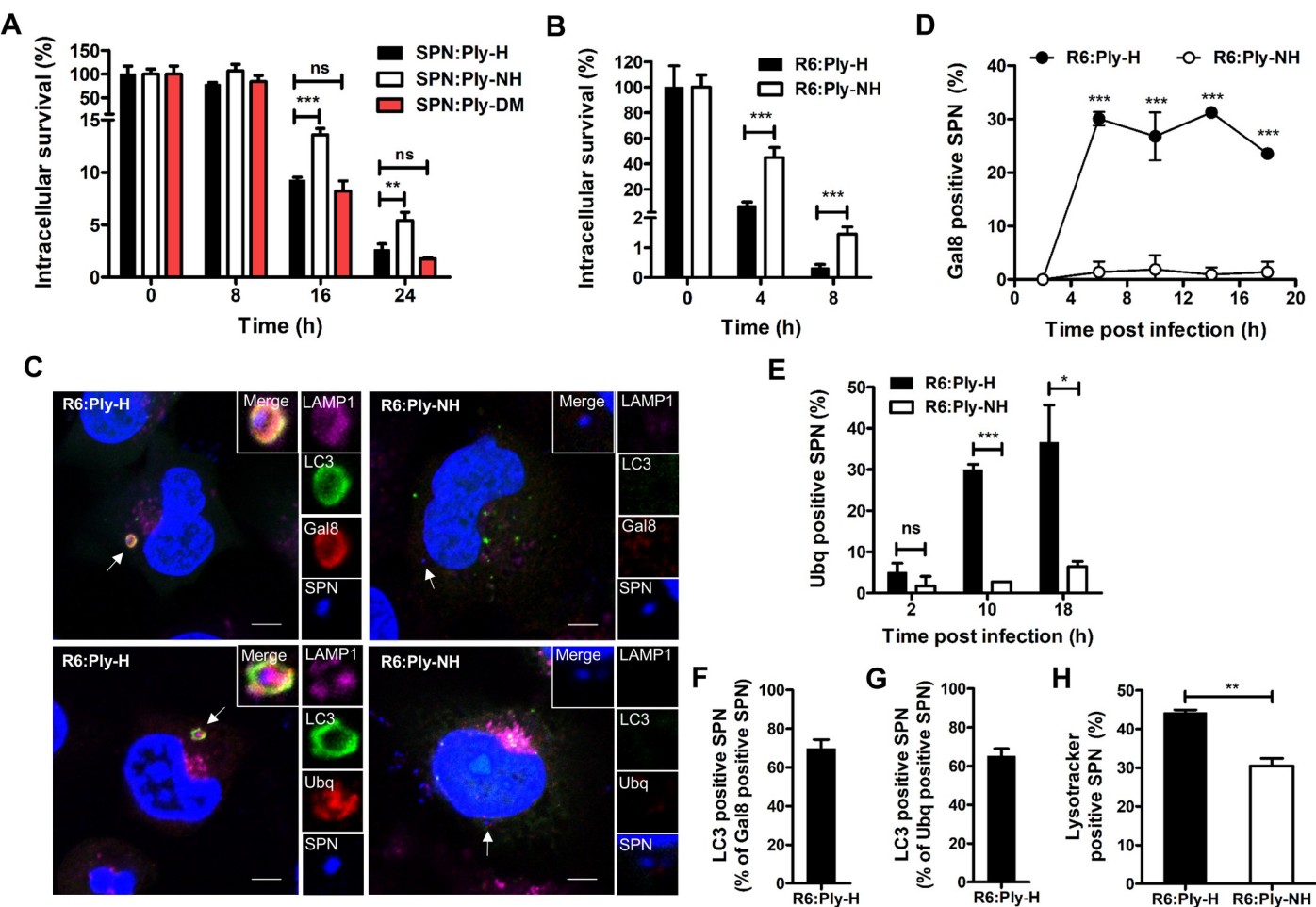

**Fig 5. Loss of pore forming ability facilitates prolonged intracellular persistence of SPN.** (**A**) Intracellular survival efficiency of SPN R6 strains expressing either Ply-H, Ply-NH or Ply-DM (H150Y+T172I) in A549s were calculated as percentage survival at indicated time points relative to 0 h. (**B**) Intracellular survival efficiency of SPN R6 strains expressing either Ply-H or Ply-NH in THP-1 macrophages. (**C**) Confocal micrographs showing association of SPN (blue) expressing Ply-H or Ply-NH with Gal8 or Ubq (red), LC3 (green) and LAMP1 (pink) in A549s at 10 h.p.i. DAPI (blue) has been used to stain A549 nucleus and SPN DNA. Arrows designate the bacteria shown in insets. Event was localized at z-stack number: 4 out of 7 (for R6:Ply-H_Gal8 image), 4 out of 7 (for R6:Ply-H_Ubq image), 4 out of 8 (for R6:Ply-NH_Gal8 image) and 7 out of 11 (for R6:Ply-NH_Ubq image). Scale bar: 5 μm. (**D**) Percent co-localization of Gal8 with SPN strains expressing either Ply-H or Ply-NH in A549s at indicated time points post-infection. (**E**) Percentage co-localization of Ubq with SPN strains expressing either Ply-H or Ply-NH in A549s at indicated time points post-infection. (**F-G**) Quantification of co-localization of LC3 with Gal8 (F) or Ubq (G) positive R6:Ply-H in A549s at 10 h.p.i. (**H**) Quantification of co-localization of SPN strains expressing either Ply-H or Ply-NH with Lysotracker in A549s at 18 h.p.i. Data information: Experiments are performed thrice and data of representative experiments are presented as mean ± SD of triplicate wells. n≥100 SPN per coverslip (D-H). Statistical analysis was performed using one-way ANOVA with Tukey's multiple comparison test (A) and Student's two-tailed unpaired t-test (B, D-E, H). ns, non-significant; *$p < 0.05$; **$p < 0.01$; ***$p < 0.001$.

that progressed in severity up until time of death, ST306 infected mice displayed minimal disease symptoms throughout (S10A Fig). This corroborates the reports of association of ST306 with respiratory tract infections that are non-lethal in nature [34,35] and previous studies in mice demonstrating attenuated virulence in ST306 isolates [35].

To determine the contribution of Ply to these phenotypes, we infected mice with either D39:Ply-NH (in which the *ply* gene had been replaced by the allele 5 *ply* from ST306), *ply* deletion strain D39Δ*ply* or D39:Ply$^{W433F}$. All strains tested were significantly attenuated in virulence as compared to D39:Ply-H (Fig 6A). The median survival times of ST306 (>168 h), D39:Ply-NH (132 h), D39Δ*ply* (>168 h) and D39:Ply$^{W433F}$ (>168 h) infected mice were significantly higher than that of D39:Ply-H (32 h) infected animals. Although 50% of D39:Ply-NH

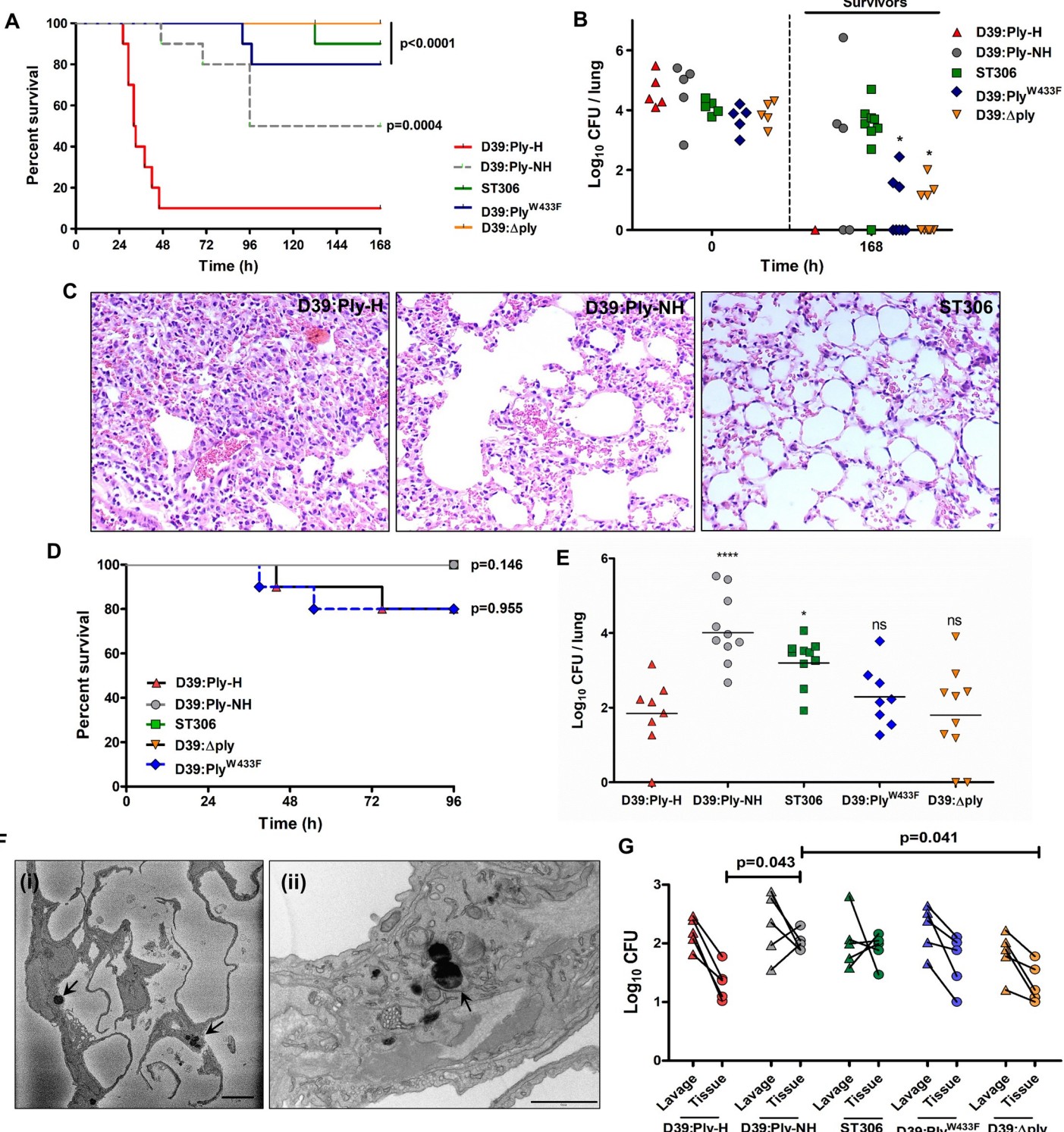

**Fig 6. Ply-NH attenuates virulence of SPN and enables host tolerance and an intracellular lifestyle in the lower respiratory tract.** (**A**) Survival time (h) of CD1 mice infected intra-nasally with $1.5\times10^6$ CFU of D39:Ply-H, D39:Ply-NH, D39:ΔPly (Ply deletion mutant), D39:Ply$^{W433F}$ (mutant strain expressing a non-hemolytic Ply) or serotype 1 ST306. n = 10 mice per group. *p*-values are from Log-rank test vs D39:Ply-H, with Bonferroni correction applied for multiple comparisons. (**B**) CFU in lungs immediately following infection (0 h, n = 5 mice per group) and in surviving mice from (**A**) at 168 h.p.i. *p*-values are from two-way ANOVA analysis with Dunnett's multiple comparisons test vs D39:Ply-NH. (**C**) Histopathology of H&E stained lung tissue samples of mice infected with D39:Ply-H, D39:Ply-NH, or ST306 SPN strains at 24 h.p.i. Magnification: 20X. (**D**) Survival time (h) of CBA/Ca mice infected intra-nasally with $1\times10^6$ CFU of D39:Ply-H, D39:Ply-NH, D39:ΔPly, D39:

Ply$^{W433}$ or ST306. n = 10 mice per group. *p*-values are from Log-rank test vs D39:Ply-H, with Bonferroni correction applied for multiple comparisons. (**E**) Lung CFU at 96 h.p.i in surviving mice from the experiment performed in (**D**). Statistical analysis was performed using one-way ANOVA with Tukey's multiple comparisons test. (**F**) (i) SBF-SEM and (ii) TEM images of mice lung sections following infection with ST306 for 24 h showing its intracellular localization (marked with arrow). Scale bar: 5 μm for (i) and 1 μm for (ii). (**G**) SPN CFU in bronchoalveolar lavage or post-lavage lung homogenates (tissue) of CBA/Ca mice at 48 h.p.i. For lavage, trachea were exposed and a syringe was used to perform 5×1 ml PBS with 1 mM EDTA washes of the lung. Pooled lavage was serially diluted onto blood agar to determine planktonic and weakly adhered pneumococcal CFU. Lavaged lungs were removed, homogenized and plated onto blood agar to determine intracellular or strongly-adherent pneumococcal CFU. The lines depict SPN CFU counts in bronchoalevolar lavage and post lavage lung homogenates from the same animal. *p*-values are from two-way ANOVA analysis with Tukey's multiple comparisons test. Experiments in (A-G) were performed a single time, with the stated numbers of animals per experimental group.

infected mice developed disease, signs of ill health appeared more slowly in these animals as compared to those infected with D39:Ply-H (S10A Fig). To determine whether the improved survival of mice was due to enhanced bacterial clearance and/or reduced bacterial growth, we determined total lung bacterial burdens in mice that survived until the end of the one week experiment and observed complete SPN clearance in the single D39:Ply-H mouse that survived (Fig 6B). By contrast, the lungs of 3/5 and 8/9 mice infected with D39:Ply-NH and ST306, respectively, still harbored moderate CFU burdens at this time point (Fig 6B). Pneumococci were recovered from lungs of 3/8 D39:Ply$^{W433F}$-infected mice and 4/10 D39Δ*ply* infected-mice at 7 days post-infection (Fig 6B). Mean recovered CFU were significantly lower in D39: Ply$^{W433F}$ (*p* = 0.0134) and D39Δ*ply* (*p* = 0.0057) infected-mice, as compared to those infected with D39:Ply-NH (Fig 6B). Although D39:Ply-NH was less virulent and demonstrated persistence relative to D39:Ply-H, it did not fully recapitulate the phenotype of ST306, indicating the involvement of other serotype/strain specific factors in the host-pathogen interactions that underpin virulence. This is demonstrated by the significant difference in survival between mice infected with D39:Ply-NH and those infected with ST306 (*p* = 0.0442, Fisher's exact test) (Fig 6A). Furthermore, in this model, D39:Ply-NH was cleared more slowly in surviving mice than either D39Δ*ply* or D39:Ply$^{W433F}$, suggesting allele 5 Ply influences infection processes in a way that is not reproduced by either loss of Ply or loss of hemolytic activity.

Since Ply is a key inducer of host inflammation, tissue injury, morbidity and mortality, we analyzed the inflammatory response in the respiratory tract of mice infected with different SPN strains. Consistent with the findings of others [36], mice infected with D39:Ply-NH and ST306 had significantly reduced lung inflammation, evident by lower expression of pro-inflammatory cytokines KC and interleukin-6 (S10C and S10D Fig) and reduced infiltration of leukocytes (S10E Fig), principally neutrophils (PMN) (S10F Fig), compared to D39:Ply-H infected animals. Histo-pathological analysis of lung sections from infected mice revealed similar trends (Fig 6C).

The introduction of high numbers of Ply-H expressing pneumococci into the murine lung triggered strong inflammatory responses which led to the death of most animals within the first 48 h, thus preventing an unbiased time point analysis of D39:Ply-H infected mice with other groups (Figs 6C and S10C–S10F). Therefore, we next adopted a murine persistence infection model, modified from that of Haste *et al.* [37], which has been shown to induce only low-level, localized inflammation within the lung instead of the acute inflammatory responses exhibited by the earlier model. Using this new model, we compared mouse survival (Fig 6D) and bacterial persistence (Fig 6E) of SPN strains expressing the Ply variants. Disease signs and premature mortality was observed in only a small proportion of those mice infected with D39: Ply-H (2/10) or D39:Ply$^{W433F}$ (2/10), whilst all mice infected with D39:Ply-NH, D39Δ*ply* or ST306 survived (Fig 6D). At four days post infection, total lung CFU was determined in surviving animals (Fig 6E). We recovered viable SPN from the lungs of all mice infected with D39: Ply-NH, ST306 or D39:Ply$^{W433F}$ and 7/8 and 8/10 mice infected with D39:Ply-H or D39Δ*ply*, respectively. The highest burdens of lung pneumococci were present in mice infected with

D39:Ply-NH or ST306 (Fig 6E), further suggesting that Ply-NH confers an advantage in lung colonization that is not fully explained either by loss of Ply-induced inflammatory responses or loss of hemolytic activity. Taken together, these results suggest that Ply-NH, aided by loss of its pore forming ability, promotes longer persistence of SPN in the lower respiratory tract and a relatively benign association with the host.

## ST306 favors an intracellular niche in the lower respiratory tract

The comparatively innocuous association of Ply-NH harboring SPN in the host lower respiratory tract was further analyzed by electron microscopic analysis (SBF-SEM and TEM) of lung sections from infected mice. For this experiment, we used the acute infection model, administering $1.5 \times 10^6$ CFU in 50 μl saline, to maximize dispersal of pneumococci throughout the lung. Intracellular ST306 and D39:Ply-NH were observed throughout lungs of mice harvested at 24 h.p.i, in SEM analysis of multiple serial block face sections (Figs 6F, S11 and S12B and S1 Movie). None were observed in D39:Ply-H infected mouse lungs and no comparable structures were seen in the uninfected control. This finding supports our *in vitro* observations that Ply-NH harboring SPN can adopt an intracellular lifestyle. We further compared SPN numbers in lung lavages (containing planktonic or weakly adherent SPN) to those in post-lavage lung tissue homogenates (containing intracellular or strongly adherent SPN) in the bacterial persistence model. At 2 days post-infection (chosen since total lung CFU in all groups were comparable at this time point; mean ± SD ($\log_{10}$): 3.52±0.48 for D39:Ply-H, 4.33±0.51 for D39: Ply-NH, 3.95±0.22 for ST306, 3.93±0.79 for D39:Ply$^{W433F}$ and 3.15±0.65 for D39Δ*ply*), 2/5 D39:Ply-NH infected mice and 3/5 ST306 infected mice had a higher number of bacteria in post-lavage tissue homogenates than in the lavage itself (Fig 6G). By contrast, all mice infected with D39:Ply-H, D39:Ply$^{W433F}$ or D39Δ*ply* had higher numbers of bacteria in lavage than in post-lavage lung homogenates.

## Discussion

One outcome of arms race driven host-pathogen interactions is Pyrrhic victory for the host, with its defence mechanisms eliminating the pathogen at the cost of inflicting lethal injury to itself. From the pathogen's perspective, this represents an infection bottleneck, limiting transmission, a problem particularly acute for obligate symbionts—such as SPN–that lack an alternate reservoir. The potential dead end that this mode of interaction presents, however, can drive co-evolution of pathogen and host into a stalemate like situation. This is referred as "disease tolerance", wherein the microbe modulates its virulence to persist in a healthy host, whilst limiting the antagonistic response being mounted against it [2]. The interaction of SPN ST306 with the host observed in our study fits this description, wherein mice infected with SPN ST306 showed prolonged survival in acute infection, despite residual bacterial burden in the lungs. The phenotype was also recapitulated, albeit to a lesser extent, with mice infected with D39:Ply-NH, which demonstrated extended survival versus those infected with D39:Ply-H. Furthermore, in a lower virulence persistence infection model, Ply-NH harbouring strains (D39:Ply-NH and ST306) colonised the lungs at a higher density than those expressing Ply-H.

Five out of six mutations that distinguish Ply-NH from Ply-H are also present in *ply* allele 3 (found in serotype 8 ST404 and ST944 isolates) that exhibits residual hemolytic activity (~1.8%) [15], implying that this variant retains its pore forming ability. However, the additional substitution of the Y150 residue (conserved across all CDCs) with histidine, is unique to Ply-NH and is primarily responsible for its complete loss of pore forming ability. Interestingly, our data indicate that Y150 engages in a novel cation-π interaction with lysine (K288) in Ply-H to facilitate pore formation, contrary to the established π-π interaction demonstrated in other

CDCs [27]. Hence, such toxins (e.g., pneumolysin and mitilysin) should be considered as a new class of CDCs (S1 Fig). Loss of this essential cation-π interaction owing to Y150H substitution, combined with the conformational rigidity of domain D3 imparted by I172, were found to be the major contributors to the loss of pore forming ability of Ply-NH. Consistent with this, a single Y150A substitution in Ply-H has been reported to reduce its hemolytic activity to 0.2% [38]. It is intriguing that subtle alterations introduced by the point mutations are sufficient to effectively influence the intra and inter-molecular interactions, rendering Ply-NH non-functional. From an evolutionary perspective, this is probably more favourable, as new traits can emerge in a short span without inflicting gross changes in the protein structure.

These minor changes in Ply sequence had a profound effect on the lifestyle of SPN. Electron microscopic analysis of mouse lung sections revealed presence of bacteria inside host cells. Comparison of SPN CFU in lung lavages to that in post-lavage lung tissue homogenates also suggested an increased propensity of Ply-NH containing SPN to embrace an intracellular life. This was aided by a significantly improved capability of Ply-NH harbouring SPN to invade host cells, which was abrogated upon treatment with Ply-H or Mβ-CD, an inhibitor of lipid raft mediated endocytosis, suggesting a role for Ply-H in the disruption of lipid raft dependent host endocytic pathways. Indeed Ply-H, but not Ply-NH, reduced the cellular internalization of cholera toxin coated latex beads, which enter cells specifically through a lipid raft mediated pathway. Recent studies have described increased affinity of Ply towards cholesterol rich domains [39] and calcium signalling triggered shedding of damaged membrane patches as a repair mechanism for Ply-created pores on plasma membranes [40]. Integrating our findings with these reports, we propose that the attempt of SPN to enter host cells is thwarted by Ply-H in its vicinity that forms pores on lipid rafts. This elicits shedding of the damaged membrane section, preventing pneumococcal entry via the lipid raft mediated pathway. Accordingly, a Ply deletion mutant in the background of strain TIGR4 has been reported to demonstrate improved internalization into Detroit 562 epithelial cells compared to its wildtype counterpart [41]. Although we believe lipid raft mediated endocytic pathways to be the key driver of the observed phenotype, pleotropic effects brought about by Ply mediated plasma membrane permeabilization, such as ion dysregulation or impairment of cell signalling could also contribute to the reduced host uptake of SPN:Ply-H. Following internalization, pore forming toxins or bacterial secretion systems can damage the bacteria-containing vacuole, resulting in bacterial clearance via induction of autophagy that sequesters damaged bacteria-containing vacuoles in double membrane structures and fuses them with lysosomes [29]. Our study demonstrates that loss of pore forming ability of Ply-NH provides an intracellular survival advantage to SPN, by enabling it to evade anti-bacterial autophagy. Additional factors that sustain a prolonged life of SPN inside intact endocytic vacuoles, however, remain elusive and require further investigation.

Thus, abrogation of Ply pore forming ability not only confers improved internalization but also ensures safe residence of SPN inside host cells. This is contrary to the general notion of SPN as an extracellular pathogen, but consistent with recent observations of its intracellular replication inside splenic macrophages and cardiomyocytes [42,43]. It is interesting to note that Ply-NH harbouring strains (D39:Ply-NH and ST306) demonstrated an added advantage in persistence in the lungs over D39Δ*ply*, in our *in vivo* mice infection model, suggesting a contribution of Ply-NH to the persistence phenotype that is not explained solely by the loss of its pore forming ability. A recent study elucidated a role of Ply in inhibiting the inflammatory responses, mediated by its interaction with MRC-1, abundant in dendritic cells and alveolar macrophages [7]. At lower doses of infection, SPN:Ply-H elicited reduced secretion of TNF-α, IL-1β and IL-12 compared to SPNΔ*ply* [7]. A non-hemolytic serotype 1 ST306 strain exhibited higher cytokine release than a closely related hemolytic serotype 1 ST228 strain [7]. Our

comparison between a SPN:Ply-NH and SPNΔ*ply* in an isogenic background revealed that the former elicits lower TNF-α from dendritic cells compared to the latter (S12A Fig), although levels in both were lower than those seen when a fully hemolytic Ply producing strain was used in the same R6 background as in Subramanian *et al* [7]. This suggests that Ply-NH retains some of its ability to bind to MRC-1 (albeit lower than Ply-H), which is supported by the fact that domain 4 of Ply was sufficient for interaction with MRC-1 [7], which is intact in Ply-NH. Furthermore, internalization of SPN via Ply-MRC-1 interaction was shown to prevent fusion of SPN containing vacuoles with the lysosomes, conferring improved intracellular survival in phagocytes [7]. Overall, these data indicate that although both SPN:Ply-NH and SPNΔ*ply* evade autophagy and demonstrate improved intracellular persistence (compared to SPN:Ply-H), the former can have an added advantage over the latter due to residual MRC-1 interaction, leading to suppression of inflammatory response and evasion of lysosomal fusion.

Collectively, our findings suggest that Ply-NH, aided partly by loss of its pore forming ability, enables host tolerance of SPN by minimising inflammation and thereby damage to both pathogen and host. Additionally, it promotes an intracellular lifestyle for SPN, enabling it to explore novel niches inside the host (Fig 7). This latter feature is likely an adaptation to the selection pressures exerted by host defences, amplified by manmade interventions in the form of vaccinations and antibiotics [44]. Although SPN:Ply-NH demonstrates improved intracellular survival (compared to SPN:Ply-H) inside both alveolar epithelial cells and THP-1 macrophages, the non-phagocytic cells of the lower respiratory tract may be a favoured niche, as the internalized SPN can establish prolonged residence and egress out of the cells with greater efficiency (S13 Fig). The need for an alternate niche is particularly relevant in the context of serotype 1 SPN, as unlike other serotypes, it has a limited asymptomatic carriage phase in the nasopharynx, the primary biological reservoir of SPN [45]. The "virulence trade-off" theory suggests that a pathogen can continue to maintain virulence as long as its transmission is uncompromised [46]. Possession of Ply-H thus presents a virulence-transmission conundrum to SPN, as Ply-H mediated inflammation is key for host to host transmission [14]. This likely explains the retention of Ply-H in other serotypes, which have a significant nasopharyngeal colonization phase in their lifecycle and a mode of transmission dependent on it. For these serotypes, progression to invasive disease and the associated risk of host mortality would be a dead-end for transmission [3]. Serotype 1 isolates harbouring Ply-NH cannot afford to be completely avirulent, as the mild respiratory tract infections they cause are likely essential to propel SPN transmission, probably in the form of coughs and sneezes, in the absence of Ply-induced inflammation. Studies indicating clonal expansion of ST306 within serotype 1 and its notable association with disease outbreaks [15] confirm that the loss of pore forming ability of Ply has not interfered with their transmission ability. Indeed, the association with outbreaks may imply that the primary transmission route is disease-dependent, rather than occurring during asymptomatic nasopharyngeal carriage. It would also be interesting to explore if ST306 adopts an intracellular life in the nasopharyngeal epithelium which would make its detection difficult in nasal swabs or washes, thus making the low prevalence of ST306 carriage an artefact of sampling methods.

Nasopharyngeal colonization that precedes invasive disease also provides the platform for intra and inter species interaction of SPN, thereby enabling acquisition and incorporation of genetic material via recombination to expand its virulence repertoire [47,48]. Given that recombination rather than mutation is the major force driving evolution of adaptive traits in SPN [49], defects in colonization and transformation might confer genetic stability, a commensal-like feature. Indeed, serotype 1 SPN are characterized by a rare, or short colonization phase [35] and poor transformability [50,51], thereby limiting avenues for expansion of genetic diversity, evident by lack of antibiotic resistance traits [52]. Additionally, Lineage A of serotype

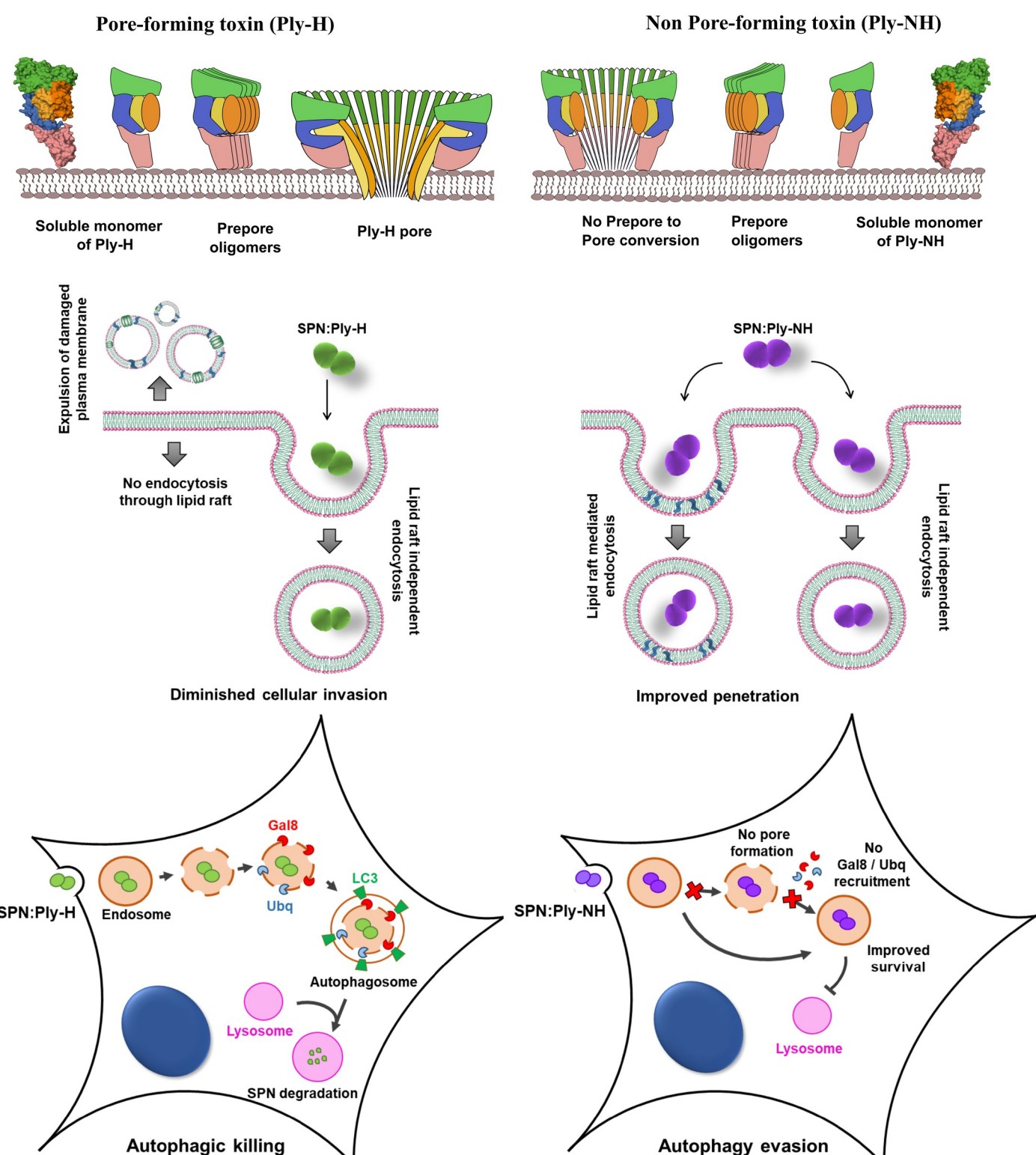

**Fig 7. Schematic of pore formation by Ply variants and its role in pneumococcal lifestyle.** Cartoon representation of major stages of pore formation by Ply-H and Ply-NH and their contribution to cellular uptake and intracellular fate of SPN.

1 SPN harbours isolates expressing non-hemolytic Ply (belonging to ST306, ST617, ST228) or variants with reduced hemolytic activity (ST227-allele 4, ST228-allele 5, 14) [15]. These isolates in particular, are associated with high disease potential, but in contrast to most other SPN, they act as primary pathogens, infecting healthy individuals, and are usually associated with low mortality rates [34]. Integrating these with our findings prompts us to speculate that expression of allele 5 *ply-NH* might assist in a more commensal-like association of serotype 1 pneumococci with the human host (like other members of the Mitis group of Streptococci), in an alternate niche, by adopting an intracellular lifestyle.

## Materials and methods

### Ethics statement

All mouse infection work was performed at the University of Liverpool with prior approval by the UK Home Office and the University of Liverpool Ethics Committee (License Number PB6DE83DA). Mice were randomly assigned to a cage (experimental group) on arrival at the unit by staff with no role in study design. Female CD1 or CBA/Ca mice of 6–8 weeks of age (Charles River, UK) were used for infection experiments and housed in individually ventilated cages for one week to acclimatize prior to infection.

### Cell culture

Human lung alveolar carcinoma (type II pneumocyte) cell line A549 (ATCC No. CCL-185) was cultured in DMEM (HiMedia) supplemented with 10% fetal bovine serum (FBS, Gibco) at 37˚C and 5% $CO_2$. THP-1 monocytes (kind gift from Sarika Mehra, IIT Bombay, India) were cultured in RPMI 1640 (Gibco) supplemented with 10% FBS at 37˚C and 5% $CO_2$. For differentiation into macrophages, THP-1 monocytes were treated with 25 ng/ml Phorbol 12-myristate 13-acetate (PMA, Sigma) for 24 h followed by resting in fresh media for another 24 h. Primary human pulmonary alveolar epithelial cells (HPAEpiC) were obtained from ScienCell Research Laboratories (Cat. No. 3200) and were cultured in Alveolar Epithelial Cell Medium (AEpiCM, ScienCell Cat. No. 3201). Dendritic cells were differentiated from healthy donor monocytes, obtained using the EasySep Human Monocyte isolation kit from Stemcell (Cat. No. 19359). Differentiation was achieved by supplementing RPMI1640 (Thermo Fisher) + 10% FBS (Sigma) with GM-CSF (40 ng/ml) and interleukin-4 (40 ng/ml) (both Peprotech, Cat. Nos. 300–03 and 200–04) for 6 days.

### *Streptococcus pneumoniae* strains

*Streptococcus pneumoniae* strains ST306 strain 01–1956 (encapsulated, serotype 1), D39 (encapsulated, serotype 2), R6 (unencapsulated derivative of D39) and TIGR4 (encapsulated, serotype 4) were routinely grown in Todd Hewitt broth supplemented with 1.5% yeast extract (THY) at 37˚C and 5% $CO_2$. For all *in vitro* studies, R6 and its derivatives were used (unless mentioned otherwise) since capsule is reported to inhibit cellular adherence and internalization [53]. *ply* allelic variants were constructed in the genetic background of D39/R6 and not ST306, as isolates belonging to serotype 1 are generally non-transformable *in vitro* [51]. For the construction of SPN:Ply-NH, *ply* ORF of D39/R6 was replaced with *ply-NH* using a cassette containing the *ply-NH* ORF (amplified from ST306 01–1956 genome) and a spectinomycin resistance cassette (amplified from pUCSpec, gift from I Biswas, KUMC, USA) flanked by *ply* upstream and downstream regions assembled in pBSK vector. Following transformation of SPN with linearized plasmid using competence stimulating peptide 1 (CSP 1) (GenPro Biotech), recombinants were selected using spectinomycin (100 µg/ml) and gene replacement was

confirmed by sequencing, hemolysis and western blot. SPN:Ply-H was similarly constructed by replacing *ply* ORF with *ply-H* ORF and a spectinomycin resistance cassette and was used in place of SPN WT, to account for any unforeseen effects of recombination [54]. R6:Ply-DM was similarly constructed following amplification of *ply-NH*$^{H150Y+I172T}$ ORF (created by site directed mutagenesis as described later). SPNΔ*ply* was generated following transformation with a construct containing the spectinomycin resistance cassette flanked by *ply* upstream and downstream regions. SPN:Ply$^{W433F}$ was constructed as described in [31]. GFP and RFP variants of SPN strains used for quantitative immunofluorescence microscopy studies were generated by transformation with *hlpA*-GFP/tagRFP cassette (gift from J-W Veening, Univ. of Lausanne, Switzerland) and selected using chloramphenicol (4.5 µg/ml), followed by confirmation with fluorescence microscopy [55]. All gene replacements were verified by PCR and DNA sequence analysis of the respective gene loci. Ply expression and activity in the generated strains were checked by western blot/ELISA and hemolysis assay, respectively (S6 Fig). A list of all the SPN strains used in this study is summarized in S3 Table.

## Antibodies and reagents

Primary antibodies used for immunofluorescence microscopy were anti-Transferrin (PA3-913, Pierce), anti-FITC (31242, Invitrogen), anti-Gal8 (AF1305, R&D Systems), anti-Ubiquitin (BML-PW8100, Enzo Life Sciences), anti-LC3 (4108, CST) and anti-LAMP1 (9091, CST). Anti-serum against SPN Enolase was a gift from S Hammerschmidt (University of Greifswald, Germany). Lysotracker Deep Red (L12492, Invitrogen), a fluorescent acidotropic dye was used to label lysosomes. For Ply western blot and ELISA, sc-80500 (Santacruz) and ab71811, ab71810 (Abcam), respectively, were used. Flow cytometry of mouse lung cells was performed with antibodies from Biolegend, including unconjugated anti CD16/32 (Fc-block) (156603), anti-CD45 FITC (103107), anti-Ly6G/Ly-6C (Gr1) PE/Cy7 (108415) and anti-F4/80 APC (123115). Phorbol 12-myristate 13-acetate (P8139), human transferrin (T8158), cholera toxin B subunit (CtxB)-FITC conjugate (C1655), methyl-β-cyclodextrin (Mβ-CD, C4555) and chlorpromazine hydrochloride (CPZ, C8138) were procured from Sigma. Human TNF-α was detected by sandwich ELISA (ab181421, Abcam).

## Protein expression and purification

The genes encoding Ply-H and Ply-NH were amplified from the genomic DNA of TIGR4 and ST306 01–1956, respectively and cloned into NdeI and XhoI sites of pET28a vector. Variants of Ply were generated by site-directed mutagenesis using respective plasmids as templates and appropriate primers (S4 Table). Ply variant constructs as well as their mutants were verified by DNA sequencing. Recombinant plasmids encoding Ply-H and Ply-NH with N-terminal His-tag were transformed into *E.coli* BL21 (DE3) cells for protein expression. Freshly transformed colonies were grown in Luria Bertani (LB) broth containing 50 µg/ml kanamycin at 37˚C on a shaker incubator for 12 h. 1% of the primary culture was added to l L of LB broth and incubated at 37˚C on a shaker incubator till the OD$_{600nm}$ reached between 0.6–0.8. Protein expression was induced by the addition of 400 µM isopropyl-1-thiogalactopyranoside (IPTG) and growing the culture further at 22˚C for 5–6 h with agitation at 150 rpm. The cells were harvested by centrifugation at 6,000 rpm for 10 min at 4˚C. The cell pellet was resuspended in buffer A (25 mM Tris, pH 8.0 and 300 mM NaCl) and lysed by sonication. Cell debris was separated by centrifugation (14,000 rpm, 50 min, 4˚C) and supernatant was applied on to a 5 ml His-Trap column, equilibrated with buffer A. The column was washed with 10 column volumes of buffer A and Ply was eluted by linear gradient of imidazole from 0 to 250 mM in buffer A. His tag was removed by treatment with TEV protease (gift from AK Varma,

ACTREC, India). Subsequently, Ply fractions were concentrated (2 ml) and applied on to a superdex-200 16/60 gel filtration column which was pre-equilibrated with buffer B (25 mM Tris, pH 8.0, 100 mM NaCl). Ply was eluted at a flow rate of 0.5 ml per min and the purity was analyzed using SDS-PAGE. The fractions containing Ply were pooled and concentrated up to 6 mg/ml using 10 kDa molecular weight cut off filter by centrifugation at 4,700 rpm at 4˚C. All the mutants of Ply were expressed and purified using this procedure. Presence of native folding in all the variants was confirmed by circular dichroism (CD) experiments.

## Crystallization of Ply-NH

The concentrated (6 mg/ml) solution of Ply-NH was used for crystallization. Initial crystallization screening was performed by sitting drop vapor diffusion method using different commercial screens at various temperatures (4, 18 and 22˚C). The needle like crystals appeared within 24 h in several conditions. Few conditions were identified for optimization. The crystal growth was further optimized by varying the precipitant and $MgCl_2$ concentration to get better diffracting quality crystals. The best quality Ply-NH crystals appeared in the crystallization drops comprised of 1 μl of Ply-NH plus 1 μl of precipitant solution containing 6 mM phosphocholine, 0.2 M $MgCl_2$ and 20% PEG 3350.

## Diffraction data collection, structure solution and refinement

The crystals were cryoprotected with mother liquor supplemented with 30% glycerol and flash frozen in liquid $N_2$ at 100 K prior to performing X-ray diffraction experiments. The complete data set was collected at BM-14 beam line at the European Synchrotron Radiation Facility (ESRF), France. A total of 300 frames, with a 1.0˚ rotation of the crystal per frame, were collected at a crystal-to-detector distance of 213 mm and with an exposure time of 8 s. Diffraction data were indexed, integrated and scaled by XDS software [56]. Systematic absence probability indicated that the crystal belonged to space group P212121 with cell dimensions a = 24.5, b = 84.7 and c = 214.6 Å. Calculation of Matthews' coefficient [57] indicated presence of one molecule in the asymmetric unit. The initial phases were obtained by molecular replacement method by program PHASER [58], using structure of Ply (PDB ID: 5CR6) as search model. Refinement of the model was performed by REFMAC5 [59] and PHENIX [60]. During the process of refinement, the manual model building was done by visual inspection of the electron density in COOT [61]. Water and other solvent molecules were added to the structure using COOT. Convergence of the refinement process was monitored by the decrease of $R_{free}$ and improvement of the overall stereochemistry. Final refinement of the complete model was performed with REFMAC5 and stereochemistry of the residues were analyzed using PRO-CHECK [62]. Data collection and refinement statistics are shown in S1 Table. All structure related figures were prepared using PyMOL [63]. The Ply-NH structure have been deposited in PDB under ID code 6JMP.

## Homology modelling of the pore-state of the CDCs

The homology models of the pore-state of Ply-NH and mitilysin were developed using the cryo-electron microscopy structure of Ply-H (PDB ID: 5LY6) [22] using the automated model building server SWISS-MODEL [64]. The sequence identity of Ply-NH and mitilysin with Ply-H are 97.87% and 97.88%, respectively. Both the generated models have good stereochemistry as they have more than 95% residues in the allowed regions of the Ramachandran plot.

## Hemolysis assay

Red blood cells (RBCs) from 1 ml of sheep blood were washed and resuspended in 50 ml PBS (pH 7.4) to get 2% (v/v) RBC suspension. 100 µl of protein samples or crude cell lysates diluted to desired concentrations in PBS were added to 96 well plate containing 50 µl DTT (10 mM) and 50 µl RBCs. After 60 min of incubation, the plates were centrifuged at 200 g for 10 min and absorbance of the supernatant was measured at 405 nm using a microplate spectrophotometer (Thermo Fischer Scientific). PBS and triton X-100 (0.05%) were used as negative and positive control, respectively.

## Western blotting

0.01 µM Ply was pre-treated with 50 µM cholesterol for 30 min at RT. 100 µl of this mixture was added to 50 µl of resealed RBC ghost membrane suspension [65] containing DTT (10 mM) and incubated for 30 min at 37˚C. The membrane was pelleted, washed 3 times in PBS and finally resuspended in 20 µl SDS loading buffer. Proteins were separated on 12% SDS-PAGE gel and following transfer to nitrocellulose membrane and incubation with anti-Ply antibody, the blot was visualized by chemiluminiscence using ECL reagent (BioRad).

To check for oligomerization, a modified protocol from Taylor *et al.* was adopted [28]. Briefly, $10^6$ A549 cells were incubated with 0.5 µg Ply variants in PBS for 30 min at 37˚C followed by addition of 2X SDS loading buffer (100 mM Tris-HCl, 200 mM DTT, 4% w/v SDS, 0.2% bromophenol blue, 20% v/v glycerol). Proteins were separated on 5% SDS-PAGE gel (without boiling) followed by immuno-blotting with anti-Ply antibody and the blots were visualized using ECL reagent.

## Liposome preparation

The liposomes were prepared by thin film hydration method [66]. Briefly, 50 mol% each of cholesterol and phosphatidylcholine (POPC) were mixed in chloroform:methanol (2:1, v/v) solution and the lipid mixture was subsequently dried in a rotary evaporator (Buchi) under vacuum for 3 h at 40˚C to form a thin film. Dried lipid layers were hydrated in buffer A (50 mM HEPES, pH 7.5 and 200 mM NaCl) and unilamellar liposomes were prepared at RT (~ 24˚C) using an extruder fitted with a polycarbonate filter of 0.2 µm pore diameter.

## Calcein leakage assay

For encapsulation of calcein in unilamellar liposomes, 20 µM of calcein was dissolved in buffer A and used for hydration of thin film as described above. The calcein dye was encapsulated by 8 freeze thaw cycles which involved freezing in liquid nitrogen for 3 min and subsequent thawing at RT (~ 24˚C) with sonication for 2 min. Calcein loaded unilamellar vesicles were prepared by extrusion as described above. The free/unencapsulated dye was separated from the liposomes by passing the extruded lipids through Sephadex G-50 column (1.5 x 50 cm) equilibrated with HEPES buffer (10 mM HEPES, 160 mM NaCl, pH 7.0). Following purification, 185 µl of protein samples diluted to desired concentration was added to 15 µl of calcein encapsulated liposomes and the fluorescence intensity of the released calcein ($\lambda_{ex}$ = 495 nm, $\lambda_{em}$ = 515 nm) was measured every min for a period of 30 min using a fluorescence spectrometer (Jasco FP-8300) maintained at 37˚C. 0.1% triton X-100 was used for complete lysis of the liposome and release of encapsulated calcein.

## TEM analysis

Freshly prepared liposomes (2 mg/ml) were incubated with purified Ply (0.5 μM) at 37˚C for 30 min, transferred to copper grids and stained with 2% phosphotungstic acid (PTA). The grids were visualized using transmission electron microscope (Philips CM200) operated at 200 kV and the images were analyzed in Image J software.

## Labelling of cysteine substituted Ply variants with NBD dye and fluorescence measurements

Cysteine free Ply variants (Ply-H[C428A] and Ply-NH[C426A]) were created by site-directed mutagenesis [67]. Two residues per TMH were selected based on the structural comparison with PFO (1PFO) to monitor the TMH1 and TMH2 formation. Residues S167, H184, D257 and E260 in Ply-H[C428A] and S175, H184, D257 and E260 in Ply-NH[C426A] were individually mutated to cysteine. These protein variants were expressed with His-tag and purified as described earlier. Following purification, 50 μM of mutant proteins were incubated with ten times higher concentration of N, N′-dimethyl-N-(iodoacetyl)-N′-(7-nitrobenz-2-oxa-1,3-diazol-4-yl) ethylenediamine 7 (NBD) at 22˚C for 2 h. After quenching the reaction with DTT (5 mM), free/unlabelled dye was separated by passing the reaction mixture through a Sephadex G-50 column pre-equilibrated with buffer B. The extent of labelling was determined by measuring the absorbance at 478 nm with extinction coefficient 25,000 $M^{-1}$ $cm^{-1}$. The cysteine substitutions and NBD labelling retained nearly >90% of the hemolytic activity in case of Ply-H. In Ply-NH, labelling with NBD also did not alter its hemolytic activity (S2 Table). NBD labelled Ply variants (200 nM) were incubated with 50 μl of liposomes (2 mg/ml) at 37˚C for 40 min to ensure oligomerization and insertion of TMHs in the membrane. All fluorescence intensity measurements ($\lambda_{ex}$ = 468 nm, $\lambda_{em}$ = 500–600 nm) of the labelled protein variants were carried out at 37˚C, in buffer B with or without incubation with liposomes, using a spectrofluorimeter (Jasco FP-8300).

## Penicillin-gentamicin protection assay

SPN strains grown until $OD_{600nm}$ 0.4 were pelleted, resuspended in PBS (pH 7.4) and diluted in assay medium for infection of A549 or primary human pulmonary alveolar epithelial cell monolayers at multiplicity of infection (MOI) of 10. Following 1 h of infection, the monolayers were washed and incubated with assay medium containing penicillin (10 μg/ml) and gentamicin (400 μg/ml) for 2 h to kill extracellular SPN. Following this, cells were lysed with 0.025% triton X-100 and the lysate was plated on Brain Heart Infusion agar to enumerate viable SPN. For infection of THP-1 macrophages, $5 \times 10^5$ cells were infected with an MOI of 1 (for invasion assay) or 0.1 (for intracellular survival assay) for 20 min followed by incubation in antibiotic containing medium for 2 h (for MOI 1) or 1 h (for MOI 0.1). Percentage invasion (internalization) was calculated as (CFU in the lysate / CFU used for infection)×100. For intracellular survival assays, the lysates were collected at different time points post antibiotic treatment and the CFU recovered were expressed as % of the CFU recovered at 0 h. The MOIs chosen were such that it did not affect the host cell viability during the entire course of the experiment (S7 Fig). For inhibition studies, A549 cells were treated before (1 h) as well as during infection with either Ply-H/Ply-NH (0.1 μg/ml) or Mβ-CD (3 and 5 mM).

## Egression assay

A549 monolayers and THP-1 macrophages were infected with SPN as described earlier at MOI of 10. Following 2 h of antibiotic treatment, the cells were incubated with 500 μl of assay

media containing 0.04 μg/ml of penicillin (bacteriostatic concentration). Every 6 h, this media was replaced and the old supernatant was spread plated on Brain Heart Infusion agar to enumerate the SPN egressed / recycled out of the cell. Percentage egression was expressed as % of the CFU recovered at 0 h.

## Dendritic cell infection

DCs were infected with SPN at MOI of 1 and extracellular bacteria were killed with 200 μg/ml of gentamicin following 2 h of infection. Culture supernatants were collected at 24 h.p.i. (22 h post-gentamicin treatment) and stored at -80˚C before quantification of cytokine production by ELISA.

## Cytotoxicity assay

Cytotoxic effect of Ply variants or SPN strains were determined by MTT assay as per manufacturer's (HiMedia) instructions.

## Preparation of FITC-cholera toxin B subunit coated latex beads

50 μl of latex bead suspension (1.1 μm, Sigma) was incubated with 25 μg of CtxB-FITC in coupling buffer (50 mM MES, pH 6.1, 200 mM NaCl) overnight at 4˚C with shaking. After washing to remove unbound toxin, coated beads were blocked in 1% bovine serum albumin (BSA) and stored in 4˚C.

## Immunofluorescence

A549 monolayers grown on coverslips were infected with SPN as described earlier at MOI of 25. For control experiments using endocytic pathway cargo, cells were incubated with transferrin (100 ng/ml) or CtxB-FITC (500 ng/ml) for 5 min and 15 min, respectively. Cells were treated before (1 h) as well as during cargo incubation with endocytic pathway specific inhibitors Mβ-CD (5 mM) or CPZ (15 μM). For latex bead uptake experiment, CtxB-FITC coated beads were added at a cell to bead ratio of 1:10 for 1.5 h.

At required time point post infection, the cells were fixed with 4% paraformaldehyde (PFA) for 15 min or methanol at -20˚C for 10 min (for ubiquitin). For PFA fixed samples, cells were permeabilized using 0.1% triton X-100 for 10 min. Following blocking in 3% BSA for 2 h, cells were treated with primary antibody for overnight at 4˚C. After treatment with secondary antibody at RT for 1 h, coverslips were finally mounted using Vectashield with DAPI (Vector laboratories). For quantitation of SPN association with autophagy/ lysosome markers, either fluorescently labelled SPN were used for infection or stained with anti-Enolase antibody postfixation. To selectively label extracellular cargo/ beads/ SPN, blocking and antibody treatments were given prior to permeabilization.

Images were acquired with an oil immersion Plan-Apochromat 63X/1.4 NA objective of a confocal laser scanning microscope (Zeiss Axio-Observer Z1). Images were acquired after optical sectioning and then processed using ZEN lite software (Version 5.0). For quantitation, at least 100 intracellular bacteria per coverslip were counted in triplicates.

## Isolation of lipid rafts by sucrose density gradient centrifugation

$5 \times 10^6$ A549 cells were collected, resuspended in 1 ml Tris buffered saline (TBS) and incubated with 25 μg of Ply variants or CtxB-FITC (+ve control, 5 μg) or transferrin (-ve control, 5 μg) in ice for 4 h. 40 μl of 25% triton X-100 was added to the mixture (final concentration 1%) and incubated in ice for 1 h. The sample was passed 20 times through $26^{1/2}$ gauge needle and

centrifuged at 10,000 g for 5 min to remove cell debris [28]. The supernatant was adjusted to 40% sucrose (in 3 ml), and overlaid with 6 ml of 30% and 3 ml of 5% sucrose (w/v) followed by centrifugation at 32,000 rpm in SW41 rotor (Beckman Coulter) for 20 h. At the end of the run, 1 ml fractions were collected from the top and 5 μl of each fraction was spotted on nitrocellulose membrane. After allowing to air-dry, the membrane was probed with either anti-Ply, anti-transferrin or anti-FITC antibody and the blot was visualized using ECL reagent (Bio-Rad).

## Mouse infections

For survival and time point pneumonia mouse experiments in acute infection model, CD1 mice were infected with $1.5 \times 10^6$ CFU of SPN in 50 μl PBS via intranasal administration under light anaesthesia with a mix of oxygen and isoflurane. For survival and time point experiments in the persistence mouse model, CBA/Ca mice were infected with $1 \times 10^6$ CFU of SPN in 30 μl PBS via intranasal administration. This protocol was adapted from that described by Haste *et al* [37]. Mice were monitored for signs of disease; pain score was determined using the scheme of Morton [68] and animals were culled at pre-determined time points or if they reached the experimental endpoint (lethargy). Lung samples were taken and homogenized using an Ultra-Turrax T8 homogenizer (IKA). Pneumococcal CFU was determined via Miles and Misra dilution onto blood agar plates containing 1 μg/ml of gentamicin. For experiments comparing lavage and tissue CFU, lungs were perfused five times with 1 ml ice-cold PBS containing 1 mM EDTA to collect lavage fluid. Lungs were then removed by dissection and processed as described above.

## Cytokine measurements (enzyme-linked immunosorbent assay)

Mouse CXCL1/KC and interleukin-6 concentrations in lung homogenates from infected mice were determined using DuoSet ELISA kits (R&D Systems, UK) according to manufacturer's instruction. A 1:5 dilution was used for all lung homogenates analyzed.

## Ply ELISA

96-well ELISA microplates (Corning Laboratories, Corning, NY) were coated overnight at 4˚C with 1 μg/well mouse anti-Ply (PLY-4) antibody (Abcam). After washing, plates were blocked for 2 h, washed again and 100 μl of bacterial lysate (prepared from $10^7$ CFU SPN from frozen stocks) was added for 2 h. After washing, 1 μg/well rabbit anti-Ply polyclonal antibody (Abcam) in 100 μl of diluent was added for 2 h. Plates were washed, and goat anti-rabbit–alkaline phosphatase antibody (Abcam) was added for 30 min. After washing, 250 μl/well pNPP color reagent (Sigma) was added for 15 min before the reaction was stopped with 50 μl of 3 N NaOH. Absorbance at 405 nm was measured with a Multiskan Spectrum microplate reader (Thermo Scientific).

## Flow cytometry

Single cell suspensions were prepared from excised mouse lungs and RBCs were lysed using RBC lysis buffer (Biolegend), according to manufacturer's instructions. Cell suspensions in PBS were incubated with purified anti-Fc receptor blocking antibody (anti-CD16/CD32) for 15 min at RT before addition of fluorochrome-conjugated antibodies against cell surface markers and incubation for 30 min at RT, in the dark. Cells were then washed and resuspended in 300 μl PBS and data acquisition performed on a Becton Dickinson FACS Canto II flow cytometer running FACSDiva acquisition software. Samples were analyzed using FlowJo software (version 8.8.3, Tree Star). Cell populations were defined as follows: leukocytes CD45

+ and neutrophils CD45+Gr-1highF4/80low/neg. The appropriate isotype control monoclonal antibodies and single conjugate controls were used to perform gating.

## Histology

Mice were euthanized 24 h after infection. Lungs were removed and fixed in 4% PFA for 24 h and changed to 70% ethanol until embedding into paraffin wax. 5 μm sections were subjected to hematoxylin and eosin staining (H&E) staining.

## Electron microscopy of mice lungs

Samples were prepared for transmission electron microscopy (TEM) and serial block face scanning electron microscopy (SBF-SEM) as follows. Mice were infected with $1.5 \times 10^6$ CFU of D39:Ply-H or ST306 and culled after 24 h. Mice were perfused with 20 ml PBS/0.1% EDTA followed by 10 ml 2.5% glutaraldehyde (w/v) in 0.1 M cacodylate buffer (pH 7.4). Whole lungs were removed, placed in fresh glutaraldehyde and fixed in a Pelco Biowave Pro (Ted Pella Inc. Redding California, USA). Tissue was further dissected into 1 mm cubes and fixed again before staining with reduced osmium (2% (w/v) $OsO_4$, 1.5% (w/v) potassium ferrocyanide in $ddH_2O$), 1% (w/v) thiocarbohydrazide (RT), 2% $OsO_4$ (w/v in $ddH_2O$), then 1% (w/v) aqueous uranyl acetate overnight at 4˚C. Next day, the tissue was finally stained with Walton's lead aspartate (0.02 M lead nitrate, 0.03 M aspartic acid, pH 5.5) at RT. To prevent precipitation artefacts, the tissue was washed copiously with $ddH_2O$ between each staining step. Unless stated, fixation and staining steps were performed in a Pelco Biowave Pro at 100w 20 Hg, for 3 min and 1 min, respectively. Dehydration was performed in a graded series of ethanol and acetone before overnight fixation and embedding in hard premix resin (TAAB, Reading, UK).

The 3View EM stack was reconstructed using Amira-Avizo Software (ThermoFisher Scientific, MA, USA). Pneumococci were rendered manually using the brush tool and volumes were calculated using the material statistics operation. Mean volume was 0.78 $\mu m^3$. Animations were made using ortho slice and animator director.

For TEM, 70–74 nm serial sections were cut using a UC6 ultra microtome (Leica Microsystems, Wetzlar, Germany) and collected on Formvar (TAAB, Reading, UK) coated Gilder 200 mesh copper grids (TAAB, Reading, UK). Images were acquired on a 120 kV Tecnai G2 Spirit BioTWIN (FEI, Hillsboro, Oregon, USA) using a MegaView III camera and analySIS software (Olympus, Germany).

## Statistical analysis

GraphPad Prism version 5 was used for statistical analysis. Statistical tests undertaken for individual experiments are mentioned in the respective figure legends. $p < 0.05$ was considered to be statistically significant. Data were tested for normality and to define the variance of each group tested. All multi-parameter analyses included corrections for multiple comparisons and data are presented as mean ± standard deviation (SD) unless otherwise stated.

## Supporting information

**S1 Fig. Evolutionary tree of the various CDCs.** The evolutionary history was inferred using the Maximum Parsimony method. The bootstrap consensus tree inferred from 500 replicates is taken to represent the evolutionary history of the taxa analyzed. Evolutionary analyses were conducted in MEGA7. Gene Bank Accession numbers: Ply-H:AAK75991.1, Ply-NH: ABO21379.1, Mitilysin:ABK58695.1, Suilysin:CAC94851.1, Intermedilysin:BAA89790.1, Vaginolysin:ACD39461.1, Inerolysin:WP_009310637.1, Ivanolysin:AQY45513.1, Listeriolysin:

CAA42639.1, Seeligeriolysin:CAA42996.1, Pyolysin:AAC45754.1, Botulinolysin:BAV54146.1, Tetanolysin:SUY56616.1, Streptolysin:NP_268546.1, Perfringolysin:WP_126964861.1, Alveo-lysin:EEL68223.1, Sphaericolysin:BAF62176.1, Cereolysin: AAX88798.1, Anthrolysin: RVU61618.1, Lectinolysin: EHE47793.1
(TIF)

**S2 Fig. Evolutionary tree of the various Ply alleles.** The evolutionary history was inferred using the Maximum Parsimony method. The bootstrap consensus tree inferred from 500 repli-cates is taken to represent the evolutionary history of the taxa analyzed. Evolutionary analyses were conducted in MEGA7. The mutations with respect to the wild type allele-1 are mentioned in parenthesis. Gene Bank Accession numbers: Allele-1:GU968409.1, Allele-2:GU968411.1, Allele-3:EF413957.1, Allele-4:EF413925.1, Allele-5:EF413960.1, Allele-6:EF413939.1, Allele-7: EF413936.1, Allele-8:GU968401.1, Allele-9:GU968397.1, Allele-10:EF413956.1, Allele-11: EF413933.1, Allele-12:EF413929.1, Allele-13:EF413924.1, Allele-15:GU968405.1, Allele-16: GU968252.1, Allele-17: GU968340.1, Allele-18: GU968232.1, Allele-19: KP982898.1, Allele-20 [17].
(TIF)

**S3 Fig. Pairwise sequence alignment of Ply-NH and Ply-H.** The amino acid sequence of Ply variants have been aligned using Clustal W. Invariant residues are highlighted in red boxes while deletion and substitutions are showed in cyan color with blue triangles. The secondary structural elements are shown for the crystal structure of Ply-NH. The figure was prepared in ESpript.
(TIF)

**S4 Fig. Electron density of map of mutations positions in Ply-NH.** Two key substitutions and one deletion region in Ply-NH. (**A**) $2F_o$-$F_c$ map (blue color) for H150. (**B**) $F_o$-$F_c$ omit map (green color) for H150. (**C**) $2F_o$-$F_c$ map (blue color) for I172. (**D**) $F_o$-$F_c$ omit map (green color) for I172. (**E**) The $2F_o$-$F_c$ map (blue color) for the loop which has the deletion.
(TIF)

**S5 Fig. Significance of cation- π interaction in pore formation by Ply-H and Mitilysin.** (**A**) Specific hemolytic activity of Ply-H mutants indicating the importance of K288 residue in the pore formation through cation-π interaction, expressed as percentage relative to Ply-H. Data is presented as mean ± SD of triplicate wells. (**B**) Structural superposition of pore-form of Ply-H and mitilysin (from *Streptococcus mitis*) demonstrating conservation of cation-π interaction. Mitilysin pore-form model was generated using Ply (5LY6) as template. The Ply-H monomers are shown in cyan and green color, while mitilysin monomers are in brown and yellow. The residue side chains are represented in ball and stick and protein molecule as cartoon.
(TIF)

**S6 Fig. Pneumolysin expression, hemolytic activity and growth rate of recombinant SPN strains used for *in vitro* and *in vivo* experiments.** (**A**) Western blot using anti-Ply and anti-Enolase (house-keeping gene) antibody to demonstrate similar expression levels of Ply across different SPN R6 strains. (**B**) Hemolysis assay of SPN R6 lysates expressed as percentage activity relative to positive control (0.05% triton X-100). (**C**) ELISA-determined Ply production per $10^7$ bacteria in D39:Ply-H, D39:Ply-NH and ST306. (**D**) Growth curves of different SPN strains measured by capturing optical density at 600 nm at different time points. Data information: Data is presented as mean ± SD of triplicate wells (B-C) or samples (D). Statistical analysis was performed using one-way ANOVA with Tukey's multiple comparison test (B-C). ns, non-sig-nificant, $^{***}p < 0.001$.
(TIF)

**S7 Fig. Cytotoxic effect associated with recombinant Ply proteins and SPN infection.** (**A**) A549 cell viability assay performed at 0 h and 24 h following infection with R6:Ply-H and R6: Ply-NH. (**B**) A549 cell viability assay performed using different concentrations of Ply-H (0.05 to 1 μg/ml) and Ply-NH. (**C**) THP-1 cell viability assay performed at 9 h following infection with indicated MOIs of R6:Ply-H and R6:Ply-NH. Data information: Uninfected cells and cells treated with 0.05% triton X-100 were taken as negative and positive controls, respectively. Experiments are performed thrice and data of representative experiments are presented as mean ± SD of triplicate wells. Statistical analysis was performed using Student's two-tailed unpaired t-test (A, C) or one-way ANOVA with Tukey's multiple comparison test (B). ns, non-significant; $^*p<0.05$; $^{**}p<0.01$; $^{***}p<0.001$.
(TIF)

**S8 Fig. Effect of different endocytic pathway specific inhibitors on uptake of cargo or CtxB-coated latex beads by A549 cells.** (**A**) Inhibition of cholera toxin B (CtxB), a lipid raft pathway specific cargo, uptake by A549 cells following treatment with Mβ-CD (5 mM, 1 h). Transferrin, a clathrin dependent endocytosis pathway specific cargo was used as negative control. Scale bar: 5 μm. (**B**) Internalization of CtxB coated latex beads by A549 cells following pre-treatment with the clathrin endocytosis inhibitor CPZ (15 μM, 1 h) and lipid raft endocytic pathway inhibitor Mβ-CD (5 mM, 1 h). Internalized beads are shown in red (arrow mark) while external beads are dual (yellow) colored. Scale bar: 5 μm.
(TIF)

**S9 Fig. Effect of Ply mediated pore formation on intracellular survival capability of SPN.** (**A**) Hemolysis assay of SPN R6 lysates expressed as percentage activity relative to positive control (0.05% triton X-100). (**B**) Percentage co-localization of Gal8 with SPN R6 strains expressing either Ply-H, Ply-NH, Δ*ply* or Ply$^{W433F}$ in A549s at 18 h.p.i. (**C**) Percentage co-localization of Ubq with SPN strains expressing either Ply-H, Ply-NH, Δ*ply* and Ply$^{W433F}$ in A549s at 18 h. p.i. (**D, E**) Intracellular survival efficiency of SPN strains expressing either Ply-H, Ply-NH and Δ*ply* mutant (**D**) or Ply-H, Ply-NH, Ply$^{W433F}$ (**E**) in A549s were calculated as percent survival at indicated time points relative to 0 h. (**F**) Intracellular survival efficiency of SPN strains expressing either Ply-H, Ply-NH or Δ*ply* mutant in THP-1s were calculated as percent survival at indicated time points relative to 0 h. Data information: Experiments are performed thrice and data of representative experiments are presented as mean ± SD of triplicate wells. Statistical analysis was performed using one-way ANOVA with Tukey's multiple comparison test (A-F). ns, non-significant; $^*p<0.05$; $^{**}p<0.01$; $^{***}p<0.001$.
(TIF)

**S10 Fig. Attenuated inflammatory responses in mice infected with Ply-NH expressing pneumococci (Acute infection model).** (**A**) Pain score, according to the scheme of Morton, in CD1 mice infected with $1.5\times10^6$ CFU SPN in 50 μl PBS. Red crosses indicate where a mouse was culled due to ill health. n = 10 mice per group. (**B**) Lung CFU over the first 2 days of infection. ELISA-determined KC (**C**) and interleukin-6 (**D**) concentrations in lung homogenates from infected mice. (**E**) Numbers of CD45+ leukocytes and (**F**) CD45+, Gr-1+, F4/80 low neutrophils (PMN) in lung homogenates from infected mice as determined by flow cytometry. *p*-values in (**C-F**) are from two-way ANOVA analysis with Dunnett's multiple comparisons test; n = 5 mice per group (B-F).
(TIF)

**S11 Fig. ST306 pneumococci show a preference for intracellular lifestyle.** (**A-F**) Electron microscopy images of ST306-infected lungs. A single mouse was infected with $1.5\times10^6$ CFU ST306 in 50 μl PBS through intranasal route and culled at 24 h.p.i. Fixative-perfused lungs

were removed and embedded in resin before imaging on a serial block face scanning electron microscope (SBF-SEM). Images in (A-F) are sequential planes 180 nm apart, covering a total depth of 1080 nm. Scale bar: 5 μm.
(TIF)

**S12 Fig. SPN:Ply-NH attenuates inflammatory response in comparison to SPNΔ*ply* and demonstrate a preference for intracellular lifestyle.** (**A**) Healthy donor dendritic cells were infected with pneumococci, at MOI 1, for 24 h and TNF-α was detected in culture supernatant by ELISA. Data are from n = 3 donors with two technical replicates per donor. *p*-values are from one-way ANOVA with Tukey's multiple comparison test. (**B**) SBF microscopy image of D39:Ply-NH-infected lung. Mouse infected with $1.5 \times 10^6$ CFU D39:Ply-NH in 50 μl PBS through intranasal route was culled at 24 h.p.i. Fixative-perfused lungs were removed and embedded in resin before imaging on a SBF-SEM. A representative image of all the images used in S1 Movie is shown. Scale bar: 5 μm.
(TIF)

**S13 Fig. Comparison of R6:Ply-NH interaction with THP-1 macrophages versus A549 alveolar epithelial cells.** (**A**) Comparison of invasion efficiency of R6:Ply-NH in THP-1 macrophages and A549 cells. (**B**) Comparison of viability of THP-1 and A549 cells performed at 0 h and 24 h following infection with R6:Ply-NH. Uninfected cells were taken as negative control (**C**) Comparison of intracellular survival efficiency of R6:Ply-NH in THP-1 and A549 cells, calculated as percent survival at indicated time points relative to 0 h. (**D**) Comparison of egression efficiency of R6:Ply-NH in THP-1 and A549 cells, calculated as percent egression at indicated time points relative to 0 h. Data information: Experiments are performed twice and data of representative experiments are presented as mean ± SD of triplicate wells. Statistical analysis was performed using Student's two-tailed unpaired t-test (A-D). ns, non-significant; $^*p < 0.05$; $^{**}p < 0.01$; $^{***}p < 0.001$.
(TIF)

**S1 Table. Diffraction data collection and structure refinement statistics.**
(DOCX)

**S2 Table. Relative hemolytic activities of cysteine substituted mutants of Ply-H and Ply-NH, before and after labelling with NBD dye.** Values shown are % hemolysis (with 100% representing complete lysis as obtained with triton X-100).
(DOCX)

**S3 Table. List of *S. pneumoniae* strains.**
(DOCX)

**S4 Table. List of primers.**
(DOCX)

**S1 Movie. 3D reconstruction of SBF-SEM images of D39:Ply-NH infected mouse lung demonstrating pneumococcal intracellular residence.** Montage made of series of serial block face images (180 nm apart) from a D39:Ply-NH infected mouse lung, showing intracellular presence of SPN. Pneumococci are colored and their 3D shape rendered from stacked images. A single section from the corresponding block is shown in S12B Fig, where the appearance of the pneumococci on the original EM image can be seen.
(MPG)

## Acknowledgments

We thank Dr. Hassan Belrhali and Dr. Babu A. Manjasetty (EMBL) for providing support on the beamline and EMBL-DBT for providing access to the BM14 beamline at the ESRF. We are thankful to Dr. Prem Prakash (IIT Bombay) for diffraction data collection. We acknowledge Sophisticated Analytical Instrumentation Facility (SAIF), IIT Bombay for the "Protein Crystallography Facility", "BSL-2 facility", "Transmission electron microscopy" and "Confocal microscopy facility". We thank Angela Platt-Higgins (Institute of Integrative Biology, University of Liverpool) for support in preparation of histology samples and acknowledge the valuable suggestions from Prof. Rodney K. Tweten (Department of Microbiology and Immunology, University of Oklahoma) regarding NBD experiment.

## Author Contributions

**Conceptualization:** Anirban Banerjee.

**Data curation:** Dilip C. Badgujar, Anjali Anil, Angharad E. Green, Daniel R. Neill, Prasenjit Bhaumik, Anirban Banerjee.

**Formal analysis:** Dilip C. Badgujar, Anjali Anil, Angharad E. Green, Manalee Vishnu Surve, Shilpa Madhavan, Daniel R. Neill, Prasenjit Bhaumik, Anirban Banerjee.

**Funding acquisition:** Daniel R. Neill, Prasenjit Bhaumik, Anirban Banerjee.

**Investigation:** Dilip C. Badgujar, Anjali Anil, Angharad E. Green, Manalee Vishnu Surve, Shilpa Madhavan, Alison Beckett, Ian A. Prior, Barsa K. Godsora, Sanket B. Patil, Prachi Kadam More, Shruti Guha Sarkar, Andrea Mitchell, Daniel R. Neill.

**Methodology:** Dilip C. Badgujar, Anjali Anil, Angharad E. Green, Manalee Vishnu Surve, Daniel R. Neill, Prasenjit Bhaumik, Anirban Banerjee.

**Project administration:** Daniel R. Neill, Prasenjit Bhaumik, Anirban Banerjee.

**Resources:** Rinti Banerjee, Prashant S. Phale, Timothy J. Mitchell, Daniel R. Neill, Prasenjit Bhaumik, Anirban Banerjee.

**Supervision:** Daniel R. Neill, Prasenjit Bhaumik, Anirban Banerjee.

**Validation:** Dilip C. Badgujar, Anjali Anil, Angharad E. Green, Daniel R. Neill, Prasenjit Bhaumik, Anirban Banerjee.

**Visualization:** Dilip C. Badgujar, Anjali Anil, Angharad E. Green, Daniel R. Neill, Prasenjit Bhaumik, Anirban Banerjee.

**Writing – original draft:** Anjali Anil, Anirban Banerjee.

**Writing – review & editing:** Dilip C. Badgujar, Anjali Anil, Angharad E. Green, Daniel R. Neill, Prasenjit Bhaumik, Anirban Banerjee.

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
