## [Decision Letter · Decision Letter 0]

7 Jul 2020

Dear Dr. Banerjee,

Thank you very much for submitting your manuscript "Structural insights into loss of function of a pore forming toxin and its role in pneumococcal adaptation to an intracellular lifestyle" for consideration at PLOS Pathogens. As with all papers reviewed by the journal, your manuscript was reviewed by members of the editorial board and by several independent reviewers. In light of the reviews (below this email), we would like to invite the resubmission of a significantly-revised version that takes into account the reviewers' comments.

This study, a new re-submission, provides a substantial amount of data, including the crystal structure, characterizing the properties of the non hemolytic PLY toxins of S. pneumoniae. The structural data are strong and the lack of pore formation despite binding and oligomerization is fascinating. All of the reviewers agree this is the major strength of the paper. Unfortunately, all of the reviewers also agree that the biological data trying to make the case for selection of this variant as a mechanism of pneumococcal persistence, are much weaker. There is concern that the data suggesting that the strains expressing non-hemolytic toxin persist intracellularly are not convincing, and the suggestion that the intracellular lifestyle is important in the disease process is not substantiated. One suggestion would be to separate this work into two papers - one with the structural data, and another to follow with a better characterization of the biological impact of this variant. Alternatively, it would be essential for the authors to address the many specific comments of the reviewers to add data substantiating their claims that the non-hemolytic toxin has a role in the complex biology of pneumococcal disease. Note, that we could not promise publication of the structural data without a resubmission of a new manuscript and review. Without a link to pathogenesis, a purely structural analysis of the toxin might be more suitable for another journal.

We cannot make any decision about publication until we have seen the revised manuscript and your response to the reviewers' comments. Your revised manuscript is also likely to be sent to reviewers for further evaluation.

Sincerely,

Alice Prince

Associate Editor

PLOS Pathogens

Michael Wessels

Section Editor

PLOS Pathogens

Kasturi Haldar

Editor-in-Chief

PLOS Pathogens

orcid.org/0000-0001-5065-158X

Michael Malim

Editor-in-Chief

PLOS Pathogens

orcid.org/0000-0002-7699-2064

This study, a new re-submission, provides a substantial amount of data, including the crystal structure, characterizing the properties of the non hemolytic PLY toxins of S. pneumoniae. The structural data are strong and the lack of pore formation despite binding and oligomerization is fascinating. All of the reviewers agree this is the major strength of the paper. Unfortunately, all of the reviewers also agree that the biological data trying to make the case for selection of this variant as a mechanism of pneumococcal persistence, are much weaker. There is concern that the data suggesting that the strains expressing non-hemolytic toxin persist intracellularly are not convincing, and the suggestion that the intracellular lifestyle is important in the disease process is not substantiated. One suggestion would be to separate this work into two papers - one with the structural data, and another to follow with a better characterization of the biological impact of this variant. Alternatively, it would be essential for the authors to address the many specific comments of the reviewers to add data substantiating their claims that the non-hemolytic toxin has a role in the complex biology of pneumococcal disease. Note, that we could not promise publication of the structural data without a resubmission of a new manuscript and review. Without a link to pathogenesis, a purely structural analysis of the toxin might be more suitable for another journal.

Reviewer's Responses to Questions

**Part I - Summary**

Reviewer #1: The authors provide a crystal structure of the pore-forming toxin, pneumolysin, from strain ST306, a strain Streptococcus pneumoniae that produces a non-hemolytic (NH) variant of pneumolysin. They go on to present data to suggest that this NH variant leads to increased phagocytosis of S. pneumoniae and facilitates intracellular persistence. In vivo, the NH variant is more rapidly cleared but also invokes a reduced inflammatory response. The study presents the new crystal structure for this non-hemolytic variant of pneumolysin, which is of interest to the field. Where this study falls short is the inconsistencies with the in vitro and in vivo data. Many of the observations seen are largely what is observed with pneumolysin knockouts or toxoid variants, and thus do not add significantly to the field. New data shows that the NH version can persist in a low dose model, the mechanism for which is not known, which would be the major shortfall of the study, when other data is limited as far as comparisons to toxoid or clean deletion strains. The major mechanistic advance would be to understand why this pneumolysin expressing versions persists better than the other variants.

Reviewer #2: The article submitted by Badgudgar et al represent a highly ambitious study on the impact of non-hemolytic pneumolysin on serotype 1 disease. The authors go on to crystalize the non-hemolytic version of pneumolysin found in serotype 1 ST306, determine that it binds to the cell, forms oligomeric rings but does not insert into the pore due to a mutation in the area that forms transmembrane b-hairpins (Y150H and T172I, that this mutation (modestly) prolongs the ability of pneumococci to survive within cells, and despite being attenuated overall for disease pneumococci that express a non-hemolytic toxin are able to persist within what appears to be an intracellular state. The conclusions reached by the authors are that this represents an evolutionary move by the bacterium to co-exist with the host as otherwise it causes death which is not optimal for the bacterium.

The authors are to be commended in what is truly a herculean effort in regards to the structural aspects of the work. Unfortunately the rigor drops off as they move towards the evidence that suggests this intracellular state is vital for the disease process and that this is a major aspect in the lifestyle of serotype 1, moreover that the ability to persist in vivo is driving type 1 disease.

One other major concern is that the authors repeatedly simplify the complexity that is pneumococcal disease. A much more nuanced statement of how pneumococcal disease arise, the role of colonization and inflammation, and disease is required. Specific concerns are detailed below.

Overall the data that suggests intracellular persistence of type 1 ST306 is vital for prolonging the bacterium state and this confers an advantage to the bacterium is insufficient. I say this not to detract form what in many ways is an excellent work effort, but to make sure that the conclusion reached by the authors are well supported and therefore find the same support by future readers. As is, I remain skeptical and I would dismiss the papers conclusions as being unsubstantiated.

Reviewer #3: The current study by Badgujar et. al concerns the pore forming toxin pneumolysin (Ply) and the structural difference between traditional Spn strain and serotype 1 isolates that are non-hemolytic. This is a resubmission of the original submission The first part of the study is based on the solved crystal structure of the NH Ply by the authors, and provides interesting contrasts with the hemolytic-Ply variants. One of the distinguishing features that makes Ply-NH unique is its ability to bind and oligomerize on cell membranes but not to form pores. Next, the authors test the differences between the two Ply in different cell culture models and establish that the loss of pore formation leads to increased internalization and persistence of Spn in host cells. Lastly, the authors test the difference in ply in a murine model, and suggest that defective NH ply strains are less virulent and persist longer in the host. The reviewer has the following concerns mentioned below

**Part II – Major Issues: Key Experiments Required for Acceptance**

Reviewer #1: Phagocytosis was performed with R6-Ply-NH and -H. How does this compare with a Ply null or toxoid derivative? The authors also mentioned differences were not due to differential pneumolysin production. Isogenically derived strains of S. pneumoniae do exist with altered Ply production, if this were to be tested.

In vivo the authors show there is improved survival of mice with the NH variant, lack of Ply is known to influence virulence. The capacity to persist was seen when bacterial counts were determined at later times (Figure 6B)-no statistics is shown in this, is NH vs H statistically different? Is this result any different to a Ply null strain? These mice no doubt have a reduced host response.

Figure 6E would appear to be an important piece of data that delineates Ply-NH vs the toxoid mutant and a Ply null strain. The question is why does this behave differently?

Reviewer #2: 1. Introduction, The authors paint a black and white picture of pneumococcal disease the bacterium is a killer. This is a simplification that is not true, the vast majority of individuals who are colonized are asymptomatic. Severe disease is in large part due to the state of the host, which is why Spn is generally considered to be an opportunistic infection. Most serotypes of Spn, the exception being type 1, have very low attack rates and do not with any frequently cause disease. Thus despite producing a hemolytic version of pneumolysin the some versions of this bacterium have adapted for long-term asymptomatic colonization. The idea that the bacteria has an attenuated toxin to reduce disease severity and prolong duration is therefore perhaps one that is restricted to serotype 1 versus being generally applicable to all of Spn.

2. Lines 80-87, Pneumolysin triggers lung epithelial cell death via necroptosis, this has been demonstrated to cause the release IL-1a from necrotic cells which is inflammatory. This should be added to the text.

3. I found the structural data to be strong and have no concerns. The authors should be commended on a very detailed and robust bit of work. The supplemental data provided by the investigator is also excellent and I congratulate the authors on their efforts to provide clarity and transparency in this manner.

4. Line 243, The authors hypothesize that it is the targeting of PLY-H on lipid rafts that is the reason why PLY-H impairs endocytotic processes whereas PLY-NH does not. Indeed PLY is likely to be found on lipid rafts as these are cholesterol rich and support the oligomerization of the toxin- ultimately causing them to be dispersed. However, it is seems far more likely that the ion-dysregulation caused by the toxin-formed pores formed by PLY-H is responsible for altering cell signaling within the cell, both due to the ion dysregulation but also energy loss in attempt to resotore the ion gradient. There is also a patch/plug response that will disrupt other normal signaling and endocytosis events – as pore-complexes are expelled. Thus the impact of hemolytic pneumolysin on a cell is very broad with lipid raft formation being only one aspect. This is not adequately recognized.

5. The results on prolonged intracellular survival in A549 cells shown in Fig5A (a difference at 16h of 9.5% to 13.5%; at 24h 2.5% to 6%) (are less than compelling). ON the other hand the data shown in Fig 5B with macrophages are more so. With data from the Oggioni lab now demonstrating that Spn replicate intracellularly and emerge, what is the fate of the bacterium and the cells that are hosting them? I think its important for the authors to consider that intracellular persistence within epithelial cells may not be why serotype1 does this and the effect may be principal in other cell types and affecting measures not examined. The fate of the hosting cells may provide an important clue.

6. Figure 6F, Given the importance of the finding to the authros claim, the authors need to provide quantitative data on the number of intacellular pneumococci they detect within infected lungs. As the authors are adept at imunofluoresect staining this requires verification with a co-stain that the bacterium are indeed intracellular.

7. Demonstration of more recoverable Spn in surviving mice infected with D39-NTH versus those that are dead is not really the same thing as showing this is an advantage for the bacterium in evolutionary manner.

Reviewer #3: 1. One the major concerns that the reviewer had was in regards to Figure 4C. Where the authors recovery rate is very few cells. The reviewer had suggested that authors carry out microscopy, which the reviewers were unable to do because of Covid-19. With such low recovery it would be beneficial to show some further data with this assay, as the ST306 is no different than WT D39. To alleviate concerns did the authors carry out this assay using an R6 with the two different variants in this assay? It would be predicted that NH would have increase intracellular numbers compared to H variant.

2. In figure 6F the authors show intracellular Spn with ST306, but none were observed with D39 H. They predict that part of the reason for that is due to NH ply variant. An important control that is missing here is the D39 NH variant, where they should be able to see intracellular bacteria.

**Part III – Minor Issues: Editorial and Data Presentation Modifications**

Reviewer #1: (No Response)

Reviewer #2: 1. Line 52. Pneumococcal genes are expressed, proteins are produced. Spn triggers inflammation by elaborating pneumolysin not expressing it.

Reviewer #3: 1. Figure 6D and G are missing any statistical analysis.

PLOS authors have the option to publish the peer review history of their article (what does this mean?). If published, this will include your full peer review and any attached files.

Reviewer #1: No

Reviewer #2: No

Reviewer #3: No
---

## [Decision Letter · Decision Letter 1]

24 Sep 2020

Dear Dr. Banerjee,

We are pleased to inform you that your manuscript 'Structural insights into loss of function of a pore forming toxin and its role in pneumococcal adaptation to an intracellular lifestyle' has been provisionally accepted for publication in PLOS Pathogens.

Best regards,

Alice Prince

Associate Editor

PLOS Pathogens

Michael Wessels

Section Editor

PLOS Pathogens

Kasturi Haldar

Editor-in-Chief

PLOS Pathogens

orcid.org/0000-0001-5065-158X

Michael Malim

Editor-in-Chief

PLOS Pathogens

orcid.org/0000-0002-7699-2064

The reviewers are satisfied with the revisions made and congratulate you on a fine study.

Reviewer Comments (if any, and for reference):

Reviewer's Responses to Questions

**Part I - Summary**

Reviewer #1: The authors have made adequate experimental attempts to address the major issues previously raised. While there are still some unanswered questions, these are beyond the scope of the existing study that presents an interesting observation on an already well defined area.

Reviewer #2: The authors have been responsive to my comments.

The text is changed as needed and the claim that reduced invasion promotes transmission is now restricted to Serotype 1.

Reviewer #3: The authors have carried out a through job of addressing this reviewers concerns raised in two rounds of reviews. As such this reviewer recommends that this manuscript be accepted.

**Part II – Major Issues: Key Experiments Required for Acceptance**

Reviewer #1: (No Response)

Reviewer #2: none

Reviewer #3: (No Response)

**Part III – Minor Issues: Editorial and Data Presentation Modifications**

Reviewer #1: (No Response)

Reviewer #2: (No Response)

Reviewer #3: In Figure 6C it would be beneficial to provide some further information on the panels (Symbols?), in the main body and also in the legend to train the reader of what to look at. According to the text in the main body there are no differences in 6C, but looking at small panel images they still look different.

PLOS authors have the option to publish the peer review history of their article (what does this mean?). If published, this will include your full peer review and any attached files.

Reviewer #1: No

Reviewer #2: **Yes: **Carlos Orihuela

Reviewer #3: No

---

## [Editor Report · Acceptance letter]

4 Nov 2020

Dear Dr. Banerjee,

We are delighted to inform you that your manuscript, "Structural insights into loss of function of a pore forming toxin and its role in pneumococcal adaptation to an intracellular lifestyle," has been formally accepted for publication in PLOS Pathogens.

Best regards,

Kasturi Haldar

Editor-in-Chief

PLOS Pathogens

orcid.org/0000-0001-5065-158X

Michael Malim

Editor-in-Chief

PLOS Pathogens

orcid.org/0000-0002-7699-2064